# Distributed Training with Heterogeneous Data: Bridging Median- and Mean-Based Algorithms

**Xiangyi Chen**[*]
University of Minnesota
chen5719@umn.edu

**Tiancong Chen**[*]
University of Minnesota
chen6271@umn.edu

**Haoran Sun**
University of Minnesota
sun00111@umn.edu

**Zhiwei Steven Wu**
Carnegie Mellon University
zstevenwu@cmu.edu

**Mingyi Hong**
University of Minnesota
mhong@umn.edu

## Abstract

Recently, there is a growing interest in the study of median-based algorithms for distributed non-convex optimization. Two prominent examples include SIGNSGD with majority vote, an effective approach for communication reduction via 1-bit compression on the local gradients, and MEDIANSGD, an algorithm recently proposed to ensure robustness against Byzantine workers. The convergence analyses for these algorithms critically rely on the assumption that all the distributed data are drawn iid from the same distribution. However, in applications such as Federated Learning, the data across different nodes or machines can be inherently heterogeneous, which violates such an iid assumption. This work analyzes SIGNSGD and MEDIANSGD in distributed settings with heterogeneous data. We show that these algorithms are non-convergent whenever there is some disparity between the expected median and mean over the local gradients. To overcome this gap, we provide a novel gradient correction mechanism that perturbs the local gradients with noise, which we show can provably close the gap between mean and median of the gradients. The proposed methods largely preserve nice properties of these median-based algorithms, such as the low per-iteration communication complexity of SIGNSGD, and further enjoy global convergence to stationary solutions. Our perturbation technique can be of independent interest when one wishes to estimate mean through a median estimator.

## 1 Introduction

In the past few years, deep neural networks have achieved great success in many tasks including computer vision and natural language processing. For many tasks in these fields, it may take weeks or even months to train a model due to the size of the model and training dataset. One practical and promising way to reduce the training time of deep neural networks is using distributed training [8]. A popular and practically successful paradigm for distributed training is the parameter server framework [16], where most of the computation is offloaded to workers in parallel and a parameter sever is used for coordinating the training process. Formally, the goal of such distributed optimization is to minimize the average of $M$ different functions from $M$ nodes,

$$\min_{x \in \mathbb{R}^d} f(x) \triangleq \frac{1}{M} \sum_{i=1}^{M} f_i(x), \tag{1}$$

where each node $i$ can only access information of its local function $f_i(\cdot)$, defined by its local data. Typically, such local objective takes the form of either a *expected* loss over local data distribution

---

[*]equal contribution

(population risk), or a *empirical average* over loss functions evaluated over a finite number of data points (empirical risk). That is,

$$f_i(x) = \int p_i(\zeta) l(x; \zeta) d\zeta, \quad \text{or} \quad f_i(x) = \frac{1}{K_i} \sum_{k=1}^{K_i} l(x; \zeta_{i,k}) \tag{2}$$

where $l(x; \zeta)$ (resp. $l(x; \zeta_{i,k})$) is the cost evaluated at a given data point $\zeta$ (resp. $\zeta_{i,k}$).

Similar to the parameter server paradigm, motivated by the use of machine learning models on mobile devices, a distributed training framework called Federated Learning has become popular [15, 18, 17]. In Federated Learning, the training data are distributed across personal devices and one wants to train a model without transmitting the users' data due to privacy concerns. While many distributed algorithms proposed for parameter server are applicable to Federated Learning, Federated Learning posed many unique challenges, including the presence of *heterogeneous* data across the nodes, and the need to accommodate asynchronous updates, as well as very limited message exchange among the nodes and the servers. By *heterogeneous* data, we mean that either $p_i(\zeta)$, or the empirical distribution formed by $\{\zeta_{i,k}\}_{k=1}^{K}$ in (2), are significantly different across the local nodes. Clearly, when the data is heterogeneous, we will have $\nabla f_i(x) \neq \nabla f_j(x)$ and if local data are *homogeneous*, we will have $\nabla f_i(x) = \nabla f_j(x)$ or $\nabla f_i(x) \approx \nabla f_j(x)$ when $K$ is large.

**Median-Based Methods.** Under these distributed optimization frameworks, many algorithms based on stochastic gradient descent (SGD) have been proposed to solve (1). The basic idea is to perform updates based on the *mean* of the local stochastic directions. On the other hand, there are two prominent and interesting algorithms whose updates are *not* directly related to the mean of the local gradients. One is called SIGNSGD (with majority vote) (see Algorithm 1) [4], which updates the parameters based on a majority vote of sign of gradients to reduce communication overheads. The other one is called MEDIANGD (see its generalized stochastic version in Algorithm 2, which we refer to as MEDIANSGD [28]), which aims to ensure robustness against Byzantine workers by using coordinate-wise median of gradients to evaluate mean of gradients.

| **Algorithm 1** SIGNSGD (with M nodes) | **Algorithm 2** MEDIANSGD (with M nodes) |
|---|---|
| 1: **Input:** learning rate $\delta$, current point $x_t$ | 1: **Input:** learning rate $\delta$, current point $x_t$ |
| 2: $g_{t,i} \leftarrow \nabla f_i(x_t) + \text{sampling noise}$ | 2: $g_{t,i} \leftarrow \nabla f_i(x_t) + \text{sampling noise}$ |
| 3: $x_{t+1} \leftarrow x_t - \delta \, \text{sign}(\sum_{i=1}^{M} \text{sign}(g_{t,i}))$ | 3: $x_{t+1} \leftarrow x_t - \delta \text{median}(\{g_{t,i}\}_{i=1}^{M})$ |

It is clear that SIGNSGD and MEDIANSGD do not simply *average* their local gradients. At first glance, their update rules also appear to be fundamentally different since they are tailored to different desiderata (that is, communication-efficiency versus robustness). Interestingly, in this work we made an observation that, SIGNSGD can be viewed as updating variables along signed median direction $(\text{sign}(\text{median}(\{g_{t,i}\}_{i=1}^{M})))$, uncovering its hidden connection to MEDIANSGD. This view provides a unified interpretation of these two algorithms as *median-based distributed algorithms*. We analyze these median-based methods in the heterogeneous regime.

**Homogeneous v.s. Heterogeneous Data.** While the median-based methods are increasingly popular, there has not been a good understanding about the convergence behavior of median-based methods. The existing analyses of both SIGNSGD and MEDIANSGD rely on the assumption of homogeneous data. SIGNSGD is analyzed from the in-sample optimization perspective: it converges to stationary points if the stochastic gradients $g_{t,i}$ sampled from each worker follow the same distribution [4, 5]. That is, $\nabla f_i(x_t) = \nabla f_j(x_t)$, $\forall x_t$, and the sampling noises follow the same distribution. On the other hand, MEDIANSGD is analyzed under the framework of population risk minimization: it converges with an optimal statistical rate, but again under the assumption that the data across the workers are iid [28].

However, in many modern distributed settings especially Federated Learning, data on different worker nodes can be inherently heterogeneous. For example, users' data stored on different worker nodes might come from different geographic regions, which induce substantially different data distributions. In Federated Learning, the stochastic gradient $g_{t,i}$ from each device is effectively the full gradient $\nabla f_i(x_t)$ evaluated on the user's data (due to the small size of local data), which violates the assumption of identical gradient distributions. Therefore, under these heterogeneous data settings, data aggregation and shuffling are often infeasible, and there is very little understanding of the behavior of both aforementioned algorithms.

From the fixed-point perspective, median-based algorithms like SIGNSGD and MEDIANSGD drive the median of gradients to 0—that is, when the median of gradients reaches 0, the algorithms will not perform updates. When the median is close to the mean of gradients (the latter is the gradient of the target loss function), it follows that the true gradient is also approximately 0, and an approximate stationary solution is reached. The reason for assuming homogeneous data in existing literature [4, 5, 28] is exactly to ensure that the median is close to mean. However, when the data from different workers are not drawn from the same distribution, the potential gap between the mean and median could prevent these algorithms from reducing the true gradient.

To illustrate this phenomenon, consider a simple one-dimensional example: $\frac{1}{3} \sum_{i=1}^{3} f_i(x) \triangleq (x - a_i)^2/2$, with $a_1 = 1, a_2 = 2, a_3 = 10$. If we run SIGNSGD and MEDIANSGD with step size $\delta = 0.001$ and initial point $x_0 = 0.0005$, both algorithms will produce iterates with large disparity between the mean and median gradients. See Fig. 1 for the trajectories of gradient norms. Both algorithms drive the median of gradients to 0 (SIGNSGD finally converges to the level of step size due to the use of sign in its update rule), while the true gradient remains a constant. In Sec. 5, we provide further empirical evaluation to demonstrate that such disparity can severely hinder the training performance.

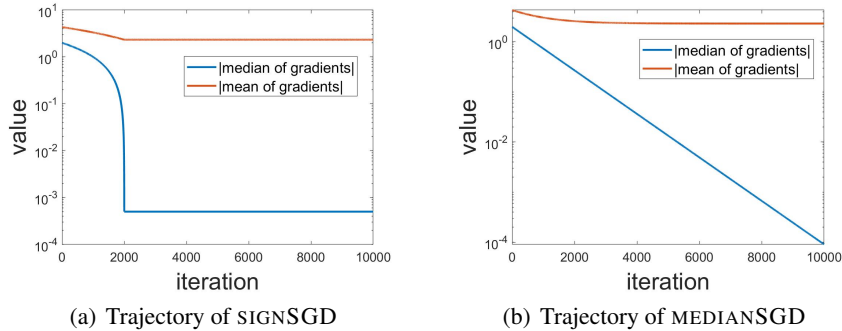

(a) Trajectory of SIGNSGD                (b) Trajectory of MEDIANSGD

Figure 1: Absolute value of mean and median of gradient vs iteration. (a) shows the trajectory of SIGNSGD (b) shows the trajectory of MEDIANSGD

**Our contribution.** Motivated by the need to understand median-based algorithms under heterogeneous data settings, we investigate two questions: 1) in a distributed training environment, under what conditions do SIGNSGD and MEDIANSGD work well? and 2) can we provide mechanisms to close the convergence gap in these algorithms? Specifically, we analyze the convergence rate of SIGNSGD and MEDIANSGD, with finite number of data samples per worker node, without assuming the data on different workers are from the same distribution. Our contributions are summarized as follows.

(1) **SIGNSGD as a median-based algorithm**: We show that SIGNSGD updates along the direction of signed *median* of gradients, which connects SIGNSGD and MEDIANSGD. This fact is crucial for our subsequent analysis of SIGNSGD, and has not been recognized by existing works so far.

(2) **Bridging the gap between median and mean by adding controlled noise.** We prove the following key result: given an arbitrary set of numbers, if one adds unimodal and symmetric noises with variance $\sigma^2$, then the expected median of the resulting numbers will approach the expected mean of the original numbers, with a rate of $O(1/\sigma)$. In addition, the distribution of the median will become increasingly symmetric as the variance of noise increases, with a rate of $O(1/\sigma^2)$. This result could be of independent interest.

(3) **Non-convergence due to the gap between median and mean.** We prove that SIGNSGD and MEDIANSGD only converge to solutions whose gradient sizes are proportional to the *difference* between the expected median and mean of gradients at different workers. Further, we show that the non-convergence is not an artifact of analysis by providing examples where SIGNSGD and MEDIANSGD does not converge due to the gap between median and mean.

(4) **Convergence of noisy SIGNSGD and noisy MEDIANSGD.** By using contribution (2), that the expected median converges to mean, and a sharp analysis on the probability density function of the noise on median, we prove that noisy SIGNSGD and noisy MEDIANSGD can both converge to stationary points.

Finally, we emphasize that the connection we established between SIGNSGD and median based method is mainly used to identify and resolve the non-convergence issue of SIGNSGD. The focus of this paper is *not* to analyze properties of median-based methods beyond convergence.

## 1.1 Related work

**Distributed training and SIGNSGD.** Distributed training of neural nets has become popular since the work of Dean et al. [8], in which distributed SGD was shown to achieve significant acceleration compared with SGD [22]. As an example, Goyal et al. [9] showed that distributed training of ResNet-50 [10] can finish within an hour. There is a recent line of work providing methods for communication reduction in distributed training, including stochastic quantization [1, 27, 26] and 1-bit gradient compression such as SIGNSGD [4, 5]. It is shown in Reddi et al. [20] that Adam [14] can diverge in some cases. Since SIGNSGD is a special case of Adam, it can suffer the same issue as Adam in general and one possible fix to this issue is by using error feedback [13]. However, it is shown in Bernstein et al. [5] that when noise is unimodal and symmetric, SIGNSGD can guarantee convergence.

**Byzantine robust optimization.** Byzantine robust optimization draws increasingly more attention in the past few years. Its goal is to ensure the performance of the optimization algorithms in the existence of Byzantine failures. Alistarh et al. [2] developed a variant of SGD based on detecting Byzantine nodes. Yin et al. [28] proposed MEDIANGD that is shown to converge with optimal statistical rate. Blanchard et al. [6] proposed a robust aggregation rule called Krum. It is shown in Bernstein et al. [5] that SIGNSGD is also robust against certain failures. Most existing works assume homogeneous data. In addition, Bagdasaryan et al. [3] showed that many existing Byzantine robust methods are vulnerable to adversarial attacks.

**Federated Learning.** Federated Learning was initially introduced in Konečný et al. [15], McMahan and Ramage [17] for collaborative training of machine learning models without transmitting users' data. It is featured by high communication costs, requirements for failure tolerance and privacy protection, as the nodes are likely to be mobile devices such as cell phones. Smith et al. [24] proposed a learning framework that incorporates multi-task learning into Federated Learning. Bonawitz et al. [7] proposed system design for large scale Federated Learning. There is a line of work on design and analysis of algorithms in federated learning includes Sattler et al. [23], Reisizadeh et al. [21], Zhou and Cong [29], Stich [25].

## 2 Distributed SIGNSGD and MEDIANSGD

In this section, we give convergence analyses of SIGNSGD and MEDIANSGD for the problem defined in (1), without any assumption on data distribution. All proofs of the results can be found in Appendix A − C. We first analyze the convergence of the algorithms under the framework of stochastic optimization. In such a setting, at iteration $t$, worker $i$ can access a stochastic gradient estimator $\hat{g}_i(x_t)$ (also denoted as $\hat{g}_{t,i}$ for simplicity). Denote the collection of the stochastic gradients to be $\{\hat{g}_t\}$. we make following assumptions thoughout the paper. A1. Unbiased gradient estimator, $\mathbb{E}[g_i(x)] = \nabla f_i(x)$. A2. Bounded variance, $\mathbb{E}[\|\text{median}(\{g_t\})_j - \mathbb{E}[\text{median}(\{g_t\})_j | x_t]\|^2] \leq \sigma_m^2$, $\forall j \in [d]$. A3. $f$ has Lipschitz gradient, i.e. $\|\nabla f(x) - \nabla f(y)\| \leq L\|x - y\|$. A4. $M$ is an odd number. The assumptions A1 and A3 are standard for stochastic optimization. A4 ensures median is well defined. A2 is a variant of classical bounded variance assumption for median-based algorithms and it is satisfied if all $\hat{g}_{t,i}$ has bounded variance.

**Notations**: Given a set of vectors $a_i$, $i = 1, ..., n$, we denote $\{a_i\}_{i=1}^n$ to be the the set and $\text{median}(\{a_i\}_{i=1}^n)$ to be the coordinate-wise median of of the vectors. We also use $\text{median}(\{a\})$ to denote $\text{median}(\{a_i\}_{i=1}^n)$ for simplicity. When $v$ is a vector and $b$ is a constant, $v \neq b$ means none of the coordinate of $v$ equals $b$. Finally, $(v)_j$ denotes $j$th coordinate of $v$, $\text{sign}(v)$ denotes the signed vector of $v$. We use $[N]$ to denote the set $\{1, 2, \cdots, N\}$.

## 2.1 Convergence of SIGNSGD and MEDIANSGD

From the pseudo-code of Algorithm 1, it is not straightforward to see how SIGNSGD is related to the median of gradients, since there is no explicit operation for calculating median in the update rule of SIGNSGD. It turns out that SIGNSGD actually goes along the signed median direction.

**Proposition 1.** *When $M$ is odd and* $\mathrm{median}(\{g_t\}) \neq 0$*, we have*

$$\mathrm{sign}(\sum_{i=1}^{M} \mathrm{sign}(g_{t,i})) = \mathrm{sign}(\mathrm{median}(\{g_t\})). \tag{3}$$

Thus, SIGNSGD updates the variables based on the sign of coordinate-wise median of gradients, while MEDIANSGD updates the variables toward the median direction of gradients. Though these two algorithms are closely related, their convergence behaviors are not well-understood. We provide the convergence guarantee for these algorithms in Theorem 1 and Theorem 2, respectively.

**Theorem 1.** *Suppose A1-A4 are satisfied, and define $D_f \triangleq f(x_1) - \min_x f(x)$. For SIGNSGD with $\delta = \frac{\sqrt{D_f}}{\sqrt{LdT}}$, the following holds true*

$$\frac{1}{T}\sum_{t=1}^{T}\mathbb{E}[\|\nabla f(x_t)\|_1] \leq \frac{3}{2}\frac{\sqrt{dLD_f}}{\sqrt{T}} + 2d\sigma_m + 2\frac{1}{T}\sum_{t=1}^{T}\mathbb{E}[\|\mathbb{E}[\mathrm{median}(\{g_t\})]|x_t] - \nabla f(x_t)\|_1]. \tag{4}$$

One key observation from Theorem 1 is that as $T$ goes to infinity, the RHS of (4) is dominated by the difference between the median and mean and the standard deviation on the median.

We remark that under the assumption that the gradient estimators from different nodes are drawn from the same unimodal and symmetric distribution in SIGNSGD, the analysis recovers the bound in Bernstein et al. [4]. In this case, we have $\mathbb{E}[\mathrm{median}(\{g_t\})|x_t] = \mathbb{E}[\nabla f(x_t)]$ and $\sigma_m = O(\sigma_l/\sqrt{M})$ if the noise on each coordinate of local gradients has variance bounded by $\sigma_l^2$ (see Theorem 1.4.1 in Miller [19]). Then, (4) becomes $\frac{1}{T}\sum_{t=1}^{T}\mathbb{E}[\|\nabla f(x_t)\|]_1 \leq \frac{3}{2}\frac{\sqrt{dLD_f}}{\sqrt{T}} + dO(\frac{\sigma_l}{\sqrt{M}})$. Under minibatch setting, one can further use a large minibatch to decrease $\sigma_l$ to show better convergence.

**Theorem 2.** *Suppose A1-A4 are satisfied, define $D_f \triangleq f(x_1) - \min_x f(x)$. Set $\delta = \min(\frac{1}{\sqrt{Td}}, \frac{1}{2L})$, MEDIANSGD yields*

$$\frac{1}{T}\sum_{t=1}^{T}\mathbb{E}[\|\nabla f(x_t)\|^2] \leq \frac{2\sqrt{d}}{\sqrt{T}}D_f + 2L\frac{\sqrt{d}}{\sqrt{T}}\sigma_m^2 + \frac{1}{T}\sum_{t=1}^{T}\mathbb{E}[\|\nabla f(x_t) - \mathbb{E}[\mathrm{median}(\{g_t\})|x_t]\|^2]. \tag{5}$$

As $T$ increases, the RHS of (5) will be dominated by the terms involving the difference between the expected median of gradients and the true gradients. In the case where the gradient from each node follows the same symmetric and unimodal distribution, the difference vanishes and the algorithm converges to a stationary point with a rate of $\frac{\sqrt{d}}{\sqrt{T}}$. However, when the gap between the expected median of gradients and the true gradients is not zero, our results suggest that both SIGNSGD and MEDIANSGD can only converge to solutions where the size of the gradient is upper bounded by some constant related to the median-mean gap.

## 2.2 Tightness of the convergence analysis

Theorem 1 − 2 suggest that it is difficult to make the *upper bounds* on the average size of the gradient of SIGNSGD and MEDIANSGD go to zero. We now provide examples to demonstrate that such a convergence gap indeed exists, thus showing that the gap in the convergence analysis is inevitable unless additional assumptions are enforced. The proof for the results below are given in Appendix D.

**Theorem 3.** *There exists a problem instance where* SIGNSGD *converges to a point $\hat{x}^*$ with*

$$\|\nabla f(\hat{x}^*)\|_1 \geq \frac{1}{T}\sum_{t=1}^{T}\mathbb{E}[\|\mathbb{E}[\mathrm{median}(\{g_t\})]] - \nabla f(x_t)\|_1] \geq 1$$

*and* MEDIANSGD *converges to*

$$\|\nabla f(\hat{x}^*)\|^2 \geq \frac{1}{T}\sum_{t=1}^{T}\mathbb{E}[\|\mathbb{E}[\mathrm{median}(\{g_t\})]] - \nabla f(x_t)\|^2] \geq 1.$$

We remark that the counter example is similar to the example for Figure 1, where the divergence is caused by the difference between median and mean. It is shown in [13] that SIGNSGD with $M = 1$ can diverge due to data sampling but we show SIGNSGD with $M \geq 1$ can diverge even if data sampling is not used. The possible divergence of MEDIANSGD is more or less noticed by the authors when developing median-based algorithms such [28, 2]. Yet, these works circumvent this issue by assuming iid data. We emphasize this non-convergence issue here because we assume non-iid data.

A traditional way to create iid data distribution in distributed training is to aggregate and shuffle the data. However, in settings with sensitive or private data such as Federated Learning, data shuffling is prohibited. This poses the question that whether it is possible to improve the performance of these median-based algorithms without transmitting data. In the following, we provide a data agnostic method to decrease the gap between median and mean, which will be used to improve median-based algorithms later.

## 3   Convergence of median towards the mean

In the previous section, we saw that there could be a convergence gap depending on the difference between expected median and mean for either SIGNSGD or MEDIANSGD. In the following, we present a general result showing that the expected median and the mean can be closer to each other if some random noise is properly added. This is the key leading to our perturbation mechanism to be proposed shortly, which ensures that SIGNSGD and MEDIANSGD can converge properly.

**Theorem 4.** *Assume we have a set of numbers $u_1, .., u_{2n+1}$. Given a symmetric and unimodal noise distribution with mean 0, variance 1. Denote the pdf of the distribution to be $h_0(z)$ and cdf to be $H_0(z)$. Suppose $h_0'(z)$ is uniformly bounded and absolutely integrable. Draw $2n + 1$ samples $\xi_1, ..., \xi_{2n+1}$ from the distribution $h_0(z)$. Define random variable $\hat{u}_i = u_i + b\xi_i$ and $\bar{u} \triangleq \sum_{i=1}^{2n+1} u_i$,*

*(a) We have*

$$\mathbb{E}[\text{median}(\{\hat{u}_i\}_{i=1}^{2n+1})] = \bar{u} + O\left(\frac{\max_{i,j}|u_i - u_j|^2}{b}\right), \tag{6}$$

$$\text{Var}(\text{median}(\{\hat{u}_i\}_{i=1}^{2n+1})) = O(b^2). \tag{7}$$

*(b) Further assume $h_0''(z)$ is uniformly bounded and absolutely integrable. Denote $r_b(z)$ to be the pdf of the distribution of $\text{median}(\{\hat{u}_i\}_{i=1}^{2n+1})$, we have*

$$r_b(\bar{u} + z) = \underbrace{\frac{1}{b}g(\frac{z}{b})}_{\text{symmetric part}} + \underbrace{\frac{1}{b}v(\frac{z}{b})}_{\text{asymmetric part}} \tag{8}$$

*where*

$$g(z) \triangleq \sum_{i=1}^{2n+1} h_0(z) \sum_{S \in \mathcal{S}_i} \prod_{j \in S} H_0(z) \prod_{k \in [n] \setminus \{i, S\}} H_0(-z) \tag{9}$$

*being the pdf of sample median of $2n + 1$ samples drawn from the distribution $h_0(z)$ which is symmetric over 0, $\mathcal{S}_i$ is the set of all $n$-combinations of items from the set $[2n + 1] \setminus i$, and the asymmetric part satisfies*

$$\int_{-\infty}^{\infty} \frac{1}{b}|v(\frac{z}{b})|dz = O\left(\frac{\max_i|\bar{u} - u_i|^2}{b^2}\right). \tag{10}$$

Eq. (6) is one key result of Theorem 4, i.e., the difference between the expected median and mean shrinks with a rate of $O(1/b)$ as $b$ grows. Another key result of the distribution is (10), which says the pdf of the expected median becomes increasingly symmetric and the asymmetric part diminishes with a rate of $O(1/b^2)$. It is worth mentioning that Gaussian distribution satisfies all the assumptions in Theorem 4. In addition, although the theorem is based on assumptions on the second-order differentiability of the pdf, we observe empirically that, many commonly used symmetric distribution with non-differentiable points such as Laplace distribution and uniform distribution can also make the pdf increasingly symmetric and make the expected median closer to the mean, as $b$ increases.

# 4 Convergence of Noisy SIGNSGD and Noisy MEDIANSGD

From Sec. 3, we see that the gap between expected median and mean will be reduced if noise is added. Meanwhile, from the analysis in Sec. 2, MEDIANSGD and SIGNSGD will finally converge to some solution whose gradient size is proportional to the above median-mean gap. Then, a natural idea is to use the perturbation mechanism to improve the performance of MEDIANSGD and SIGNSGD.

In this section, we propose and analyze the noisy variants of SIGNSGD and MEDIANSGD, where symmetric and unimodal noises are injected on the local gradients. The only difference between the noisy algorithms and the original algorithms is that some artificial noise is added to the stochastic gradients before further processing, i.e. changing line 2 of A (see Algorithm 3 and Algorithm 4 for pseudo code).

| **Algorithm 3** Noisy SIGNSGD | **Algorithm 4** Noisy MEDIANSGD |
|---|---|
| **Input:** learning rate $\delta$, current point $x_t$ | **Input:** learning rate $\delta$, current point $x_t$ |
| $g_{t,i} = \hat{g}_{t,i} + b\xi_{t,i}$ | $g_{t,i} = \hat{g}_{t,i} + b\xi_{t,i}$ |
| $x_{t+1} \leftarrow x_t - \delta \operatorname{sign}(\sum_{i=1}^{M} \operatorname{sign}(g_{t,i}))$ | $x_{t+1} \leftarrow x_t - \delta \operatorname{median}(\{g_{t,i}\}_{i=1}^{M})$ |

**Remark on the sampling noise and connection to differential privacy:** The above algorithms still follow the update rule of SIGNSGD and MEDIANSGD, just that the noise on the gradients follows some distribution with good properties described in Section 3. Essentially we want the gradient noise to be symmetric. If the noise generated by data sub-sampling is approximate symmetric, this sub-sampling noise should also help with the convergence performance. It is shown in Bernstein et al. [4] that the gradient noise generated by data sub-sampling indeed follows some symmetric structure and we show later in Section 5 that sub-sampling indeed helps in practice under the situation of heterogeneous data. One may notice the noise $\xi_{t,i}$ is similar to the noise added to ensure differential privacy when it is Gaussian. If an upper bound on gradient norm is known, one can use standard privacy accountant to calculate the privacy cost. Our convergence result implies that certain amount of noise added for differential privacy can help instead of hurt. However, the theoretical utilities of these algorithms under the privacy setting are unknown and we leave them for future works.

Now we present the convergence results for Algorithm 3 and Algorithm 4.

**Theorem 5.** *Suppose A1, A3, A4 are satisfied and $|(\hat{g}_{t,i})_j| \leq Q, \forall t, j$. When each $\xi_{t,i}$ is sampled iid from a symmetric and unimodal distribution with mean 0, variance 1 and pdf $h(z)$. If $h'(z)$ and $h''(z)$ are uniformly bounded and absolutely integrable. Define $\sigma$ to be standard deviation of median of $2n+1$ samples drawn from $h(z)$, $D_f \triangleq f(x_1) - \min_x f(x)$, $\mathcal{W}_t$ to be the set of coordinates $j$ at iteration $t$ with $\frac{|\nabla f(x_t)_j|}{b\sigma} \geq \frac{2}{\sqrt{3}}$. For SIGNSGD, we have*

$$\frac{1}{T}\sum_{t=1}^{T}\left(\sum_{j\in\mathcal{W}_t}|\nabla f(x_t)_j| + \sum_{j\in[d]\setminus\mathcal{W}_t}\frac{1}{b\sigma}\nabla f(x_t)_j^2\right) \leq \frac{3D_f}{T\delta} + \frac{3}{T}\sum_{t=1}^{T}\mathbb{E}\left[\sum_{j=1}^{d}|\nabla f(x_t)_j|O\left(\frac{1}{b^2}\right)\right] + \frac{3L}{2}\delta d \tag{11}$$

**Effect of $b$ and comparison with error feedback.** Theorem 5 shows how different parameters can affect the convergence guarantee. The term that decreases with $b$ on RHS of (11) is the convergence gap introduced by heterogeneous data and data sampling. Without artificial noise, this term is a constant. Such a term implies adding noise can provably improve the performance of SIGNSGD if it is stuck due to the median-mean gap. In the proof, this gap appears when bounding asymmetricity of distribution of the median gradient, which can decrease after adding noise and can be small on a homogeneous distribution (see (84) in the Appendix). The other two terms on the RHS of (11) are standard in convergence analysis and can diminish with proper $T$ and $\delta$. The trade-off between convergence speed and final accuracy is reflected on LHS of (11), i.e. the algorithm can converge better but slower with larger noise. If one optimizes $b$ and $\delta$ w.r.t $T$ and $d$, one can get a worst case bound $\frac{1}{T}\sum_{t=1}^{T}\left(\sum_{j\in\mathcal{W}_t}T^{1/4}d^{1/4}|\nabla f(x_t)_j| + \sum_{j\in[d]\setminus\mathcal{W}_t}\nabla f(x_t)_j^2\right) \leq O\left(\frac{d^{3/4}}{T^{1/4}}\right)$ by setting $b = T^{1/4}d^{1/4}$ and $\delta = 1/\sqrt{Td}$. Such a rate is slower than SGD's $O(\sqrt{d}/\sqrt{T})$ rate, and it implies that SIGNSGD may not be preferred when data heterogeneity is large. Compared with the error feedback fix of SIGNSGD in [13]. Our approach address the convergence issue in multi-worker setting and does not modify update rule SIGNSGD, while the error feedback fix is designed for single-worker setting and uses gradient magnitude information in parameter updates.

**Theorem 6.** *Suppose A1, A3, A4 are satisfied and $|(\hat{g}_{t,i})_j| \leq Q, \forall t, j$. When each coordinate of $\xi_{t,i}$ is sampled iid from a symmetric and unimodal distribution with mean 0, variance 1 and pdf $h(z)$. If $h'(z)$ is uniformly bounded and absolutely integrable. Define $D_f \triangleq f(x_1) - \min_x f(x)$, set $\delta \leq \frac{1}{2L}$. For* MEDIANSGD*, we have*

$$\frac{1}{T}\sum_{t=1}^{T}\mathbb{E}[\|\nabla f(x_t)\|^2] \leq \frac{2}{T\delta}D_f + O\left(\frac{d}{b^2}\right) + O\left(\delta d b^2\right) \tag{12}$$

**Effect of $b$ and robustness after perturbation.** The trade-off between convergence speed and convergence accuracy in Theorem 6 is more clear compared with Theorem 5. Larger $b$ can effectively reduce the median-mean gap and simultaneously induce a larger noise on gradient. The best convergence rate is $O(d^{2/3}/T^{1/3})$ by setting $b = T^{1/6}d^{1/6}$ and $\delta = 1/(T^{2/3}d^{2/3})$. The robustness of MEDIANSGD come from the fact that when performing variable updates, the mean of gradients is replaced by the median of gradients, which is less sensitive to extreme values. Since the noise perturbation technique is gradually converting the median estimator to a mean estimator as more noise is added, it should make MEDIANSGD less robust simultaneously. On heterogeneous data with Byzantine workers, the performance of the median-based algorithms can be affected by mainly two factors. One is the gap between median and mean of gradients, the other one is the misleading information provided by Byzantine workers. When the possible effect of Byzantine workers is small (e.g. a small number of Byzantine workers) compared with the median-mean gap, some noise might be still preferred to reduce the median-mean gap even though this could amplify the effect of Byzantine workers. For non-heterogeneous data (e.g. the data is distributed iid across workers), the median of gradient could be very close to the mean especially when the number of training samples is large. In such cases, the median-mean gap may not be a bottleneck for the performance of the algorithm and the noise scale $b$ should be set small or even 0. There could be a trade-off between convergence guarantee and robustness and a through quantitative study is left for future works.

## 5 Experiments

In this section, we show how adding noise helps the practical behavior of the algorithms. Since SIGNSGD is better studied empirically and MEDIANSGD is more of theoretical interest so far, we use SIGNSGD to demonstrate the benefit of injecting noise. We conduct experiments on MNIST and CIFAR-10 datasets. For both datasets, the data distribution on each node is heterogeneous, more specifically, each node contains some exclusive data for one or two out of ten categories. More details about the experiment configuration can be found in Appendix I.

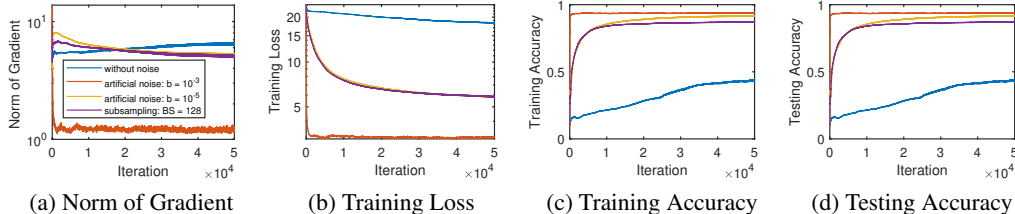

| (a) Norm of Gradient | (b) Training Loss | (c) Training Accuracy | (d) Testing Accuracy |

Figure 2: Comparison of SIGNSGD with different noise on MNIST.

We first compare full batch Noisy SIGNSGD (Algorithm 3) with different $b$ (i.e. adding random Gaussian noise with different standard deviation), SIGNSGD with sub-sampling on data , and full batch SIGNSGD without any noise  on MNIST. This experiment is to check the effects of the noise generated from data sub-sampling and the artificial noise. We see that SIGNSGD without noise stuck at some point where the gradient is a constant (the median should be oscillating around zero). At the same time, both Noisy SIGNSGD or SIGNSGD with sub-sampling drive the gradient to be smaller. From the results in Figure 2, we can see that both training accuracy and test accuracy of SIGNSGD without noise are very poor. The perturbation indeed helps improve the classification accuracy in practice.

In the second experiment (see Figure 3), we examine the performance of Noisy SIGNSGD on CIFAR-10. We compare SIGNSGD on both heterogeneous and homogeneous data, and Noisy SIGNSGD  on heterogeneous data. We can see that when artificial noise is added to the gradients, the convergence speed of SIGNSGD is significantly increased. In addition, we can see that a side benefit of adding

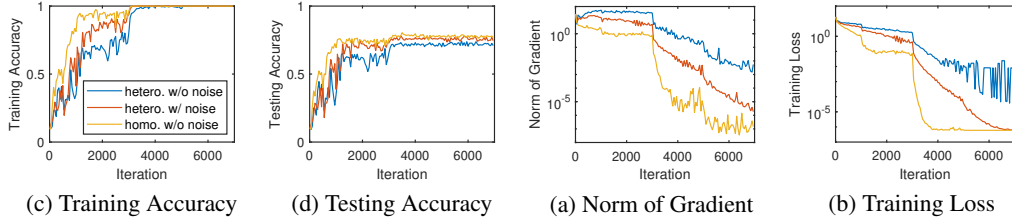

(c) Training Accuracy    (d) Testing Accuracy    (a) Norm of Gradient    (b) Training Loss

Figure 3: Comparison of SIGNSGD on CIFAR-10. For the noisy algorithms we use $b = 0.001$. The sudden change of performance is caused by learning rate decay, which happens at 1000/3000/5000 iterations.

noise is improving generalization performance, despite all the algorithm achieves almost 100% training accuracy, Noisy SIGNSGD achieves higher testing accuracy than vanilla SIGNSGD. It is shown that SIGNSGD on homogeneous data converges with similar speed of SGD in [4] while reducing 16x or 32x communication per iteration. Comparing convergence speed of Noisy SIGNSGD on heterogeneous data and SIGNSGD on homogeneous data, one can argue Noisy SIGNSGD should also be communication efficient when considering total communication compared with SGD. This set of experiments again shows that injecting noise does help in practice.

## 6    Conclusion and discussion

In this paper, we uncover the connection between SIGNSGD and MEDIANSGD by showing SIGNSGD is a median-based algorithm. We also show that when the data at different nodes come from different distributions, the class of median-based algorithms suffers from non-convergence caused by using the median to evaluate mean. To fix the non-convergence issue, We provide a perturbation mechanism to shrink the gap between the expected median and mean. By incorporating the perturbation mechanism, we show the convergence of both SIGNSGD and MEDIANSGD is guaranteed to improve. To the best of our knowledge, this is the first time that median-based methods, including SIGNSGD and MEDIANSGD, are able to converge with a provable rate for distributed problems with heterogeneous data. The perturbation mechanism can be approximately realized by sub-sampling of data during gradient evaluation, which partly supports the use of sub-sampling in practice. We also conducted experiments on training neural nets to show the necessity of the perturbation mechanism and sub-sampling.

## 7    Broader Impacts

As machine learning models are trained on increasingly large datasets, more and more training processes are done in a distributive fashion, where the training data are distributed across the nodes in a network. These nodes may correspond to data centers across different regions or mobile devices of individual users. In these settings, the data distributions across different nodes can be inherently heterogeneous. Under this setting of heterogeneous distributions across nodes, our work provides new understandings on the behavior of popular distributed training algorithms that are optimized for robustness and communication efficiency. These insights can be useful for practitioners to formally reason about the trade-offs across accuracy, robustness, and communication efficiency.

## Acknowledgments and Disclosure of Funding

The research is supported in part by grants AFOSR-19RT0424, ARO-W911NF-19-1-0247, a Google Faculty Research Award, a J.P. Morgan Faculty Award, and a Facebook Research Award.

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
