[Supplementary Material]

## A Proof of Proposition 1

Suppose we have a set of numbers $a_k, k \in [M], a_k \neq 0, \forall k$ and $M$ is odd. We show the following identity

$$\text{sign}\left(\sum_{k=1}^{M} \text{sign}(a_k)\right) = \text{sign}(\text{median}(\{a_k\}_{k=1}^{M})) \tag{13}$$

To begin with, define $b_k, k \in [M]$ to be a sequence of $a_k$ sorted in ascending order. Then we have

$$\text{median}(\{a_k\}_{k=1}^{M}) = \text{median}(\{b_k\}_{k=1}^{M}) = b_{(M+1)/2} \tag{14}$$

and the following

$$\text{sign}\left(\sum_{k=1}^{M} \text{sign}(a_k)\right) = \text{sign}\left(\sum_{k=1}^{M} \text{sign}(b_k)\right)$$

$$= \text{sign}\left(\text{sign}(b_{(M+1)/2}) + \sum_{k=1}^{(M+1)/2-1} \text{sign}(b_k) + \sum_{k=(M+1)/2+1}^{M} \text{sign}(b_k)\right). \tag{15}$$

Recall that $b_k$ is non-decreasing as it is a sorted sequence of $a_k$ with ascending order. If $b_{(M+1)/2} > 0$, we have $b_k > 0, \forall k > (M+1)/2$ and thus

$$\sum_{k=(M+1)/2+1}^{M} \text{sign}(b_k) = \sum_{k=(M+1)/2+1}^{M} 1 = (M-1)/2. \tag{16}$$

Since $\sum_{k=(M+1)/2+1}^{M} \text{sign}(b_k) \geq \sum_{k=(M+1)/2+1}^{M} -1 = -(M-1)/2$, we have

$$\text{sign}(b_{(M+1)/2}) + \sum_{k=1}^{(M+1)/2-1} \text{sign}(b_k) + \sum_{k=(M+1)/2+1}^{M} \text{sign}(b_k) \geq (\text{sign}(b_k)) = 1 \tag{17}$$

which means when $\text{median}(a_k) > 0$,

$$\text{sign}\left(\sum_{k=1}^{M} \text{sign}(a_k)\right) = 1 \tag{18}$$

Following the same procedures as above, one can also get when $\text{median}(a_k) < 0$,

$$\text{sign}\left(\sum_{k=1}^{M} \text{sign}(a_k)\right) = -1 \tag{19}$$

Thus,

$$\text{sign}\left(\sum_{k=1}^{M} \text{sign}(a_k)\right) = \text{sign}\left(\text{median}(a_k)\right) \tag{20}$$

when $\text{median}(a_k) \neq 0$.

Applying the result above to each coordinate of the gradient vectors finishes the proof. □

## B Proof of Theorem 1

Let us define:

$$\text{median}(\{g_t\}) \triangleq \text{median}(\{g_{t,i}\}_{i=1}^{M}). \tag{21}$$

and

$$\text{median}(\{\nabla f_t\}) \triangleq \text{median}(\{\nabla f_i(x_t)\}_{i=1}^M). \tag{22}$$

By A3, we have the following standard descent lemma in nonconvex optimization.

$$f(x_{t+1}) \leq f(x_t) + \langle \nabla f(x_t), x_{t+1} - x_t \rangle + \frac{L}{2}\|x_{t+1} - x_t\|^2 \tag{23}$$

Substituting the update rule into (23), we have the following series of inequalities

$$
\begin{aligned}
f(x_{t+1}) &\leq f(x_t) - \delta \langle \nabla f(x_t), \text{sign}(\text{median}(\{g_t\}))\rangle + \frac{L}{2}\delta^2 d \\
&= f(x_t) - \delta \langle \mathbb{E}[\text{median}(\{g_t\})], \text{sign}(\text{median}(\{g_t\}))\rangle \\
&\quad + \delta \langle \mathbb{E}[\text{median}(\{g_t\})] - \nabla f(x_t), \text{sign}(\text{median}(\{g_t\}))\rangle + \frac{L}{2}\delta^2 d \\
&\leq f(x_t) - \delta \|\mathbb{E}[\text{median}(\{g_t\})]\|_1 + \delta \|\mathbb{E}[\text{median}(\{g_t\})] - \nabla f(x_t)\|_1 \\
&\quad + 2\delta \sum_{j=1}^{d} |\mathbb{E}[\text{median}(\{g_t\})_j]| I[\text{sign}(\text{median}(\{g_t\})_j) \neq \text{sign}(\mathbb{E}[\text{median}(\{g_t\})_j])] \\
&\quad + \frac{L}{2}\delta^2 d \tag{24}
\end{aligned}
$$

where $\text{median}(\{g_t\})_j$ is $j$th coodrinate of $\text{median}(\{g_t\})$, and $I[\cdot]$ denotes the indicator function.

Taking expectation over all the randomness, we get

$$
\begin{aligned}
&\delta \mathbb{E}[\|\mathbb{E}[\text{median}(\{g_t\})]\|_1] \\
&\leq \mathbb{E}[f(x_t)] - \mathbb{E}[f(x_{t+1})] + \delta \mathbb{E}[\|\mathbb{E}[\text{median}(\{g_t\})]] - \nabla f(x_t)\|_1] \\
&\quad + 2\delta E\left[\sum_{j=1}^{d} |\mathbb{E}[\text{median}(\{g_t\})_j]| P[\text{sign}(\text{median}(\{g_t\})_j) \neq \text{sign}(\mathbb{E}[\text{median}(\{g_t\})_j])]\right] \\
&\quad + \frac{L}{2}\delta^2 d \tag{25}
\end{aligned}
$$

Before we proceed, we analyze the error probability of sign

$$P[\text{sign}(\text{median}(\{g_t\})_j) \neq \text{sign}(\mathbb{E}[\text{median}(\{g_t\})_j])] \tag{26}$$

This follows a similar analysis as in SIGNSGD paper.

By reparameterization, we can have

$$\text{median}(\{g_t\})_j = \mathbb{E}[\text{median}(\{g_t\})_j] + \zeta_{t,j}$$

with $\mathbb{E}[\zeta_{t,j}] = 0$.

By Markov inequality and Jensen's inequality, we have

$$
\begin{aligned}
&P[\text{sign}(\text{median}(\{g_t\})_j) \neq \text{sign}(\mathbb{E}[\text{median}(\{g_t\})_j])] \\
&\leq P[|\zeta_{t,j}| \geq \mathbb{E}[\text{median}(\{g_t\})_j]] \\
&\leq \frac{\mathbb{E}[|\zeta_{t,j}|]}{\mathbb{E}[\text{median}(\{g_t\})_j]} \\
&\leq \frac{\sqrt{\mathbb{E}[\zeta_{t,j}^2]}}{\mathbb{E}[\text{median}(\{g_t\})_j]} = \frac{\sigma_m}{\mathbb{E}[\text{median}(\{g_t\})_j]} \tag{27}
\end{aligned}
$$

where we assumed $\mathbb{E}[\zeta_{t,j}^2] \leq \sigma_m^2$.

Substitute (27) into (25), we get

$$
\begin{aligned}
&\delta \mathbb{E}[\|\mathbb{E}[\text{median}(\{g_t\})]\|_1] \\
&\leq \mathbb{E}[f(x_t)] - \mathbb{E}[f(x_{t+1})] + \delta \mathbb{E}[\|\mathbb{E}[\text{median}(\{g_t\})]] - \nabla f(x_t)\|_1] + 2\delta d\sigma_m + \frac{L}{2}\delta^2 d. \tag{28}
\end{aligned}
$$

Now we use standard approach to analyze convergence rate. Summing over $t$ from 1 to $T$ and divide both sides by $T\delta$, we get

$$\frac{1}{T}\sum_{t=1}^{T}\mathbb{E}[\|\mathbb{E}[\mathrm{median}(\{g_t\})]\|_1]$$

$$\leq \frac{D_f}{T\delta} + \frac{1}{T}\sum_{t=1}^{T}\mathbb{E}[\|\mathbb{E}[\mathrm{median}(\{g_t\})]] - \nabla f(x_t)\|_1] + 2d\sigma_m + \frac{L}{2}\delta d \qquad (29)$$

where here we defined $D_f \triangleq \mathbb{E}[f(x_1)] \min_x f(x)$

Now set $\delta = \frac{\sqrt{D_f}}{\sqrt{LdT}}$, we get

$$\frac{1}{T}\sum_{t=1}^{T}\mathbb{E}[\|\mathbb{E}[\mathrm{median}(\{g_t\})]\|_1]$$

$$\leq \frac{3}{2}\frac{\sqrt{dLD_f}}{\sqrt{T}} + \frac{1}{T}\sum_{t=1}^{T}\mathbb{E}[\|\mathbb{E}[\mathrm{median}(\{g_t\})]] - \nabla f(x_t)\|_1] + 2d\sigma_m \qquad (30)$$

Going one step further, and use the triangular inqaulity, we can easily bound the $\ell_1$ norm of the gradient as the following

$$\frac{1}{T}\sum_{t=1}^{T}\mathbb{E}[\|\nabla f(x_t)\|_1] \leq \frac{3}{2}\frac{\sqrt{dLD_f}}{\sqrt{T}} + 2\frac{1}{T}\sum_{t=1}^{T}\mathbb{E}[\|\mathbb{E}[\mathrm{median}(\{g_t\})]] - \nabla f(x_t)\|_1] + 2d\sigma_m \quad (31)$$

## C   Proof of Theorem 2

By the gradient Lipschitz continuity and the update rule, we have

$$f(x_{t+1})$$
$$\leq f(x_t) - \delta\langle\nabla f(x_t), \mathrm{median}(\{g_t\})\rangle + \frac{L}{2}\delta^2\|\mathrm{median}(\{g_t\})\|^2$$
$$\leq f(x_t) - \delta\langle\nabla f(x_t), \mathrm{median}(\{g_t\})\rangle$$
$$\quad + L\delta^2(\|\mathrm{median}(\{g_t\}) - \mathbb{E}[\mathrm{median}(\{g_t\})|x_t]\|^2 + \|\mathbb{E}[\mathrm{median}(\{g_t\})|x_t]\|^2)$$

Taking expectation, we have

$$\mathbb{E}[f(x_{t+1})] - \mathbb{E}[f(x_t)]$$
$$\leq -\delta\mathbb{E}_{x_t}[\langle\nabla f(x_t), \mathbb{E}_{\{g_t\}}[\mathrm{median}(\{g_t\})|x_t]\rangle]$$
$$\quad + L\delta^2\mathbb{E}[\|\mathrm{median}(\{g_t\}) - \mathbb{E}[\mathrm{median}(\{g_t\})|x_t]\|^2 + \|\mathbb{E}[\mathrm{median}(\{g_t\})|x_t]\|^2]$$
$$= -\delta\mathbb{E}_{x_t}\left[\frac{1}{2}\left(\|\nabla f(x_t)\|^2 + \|\mathbb{E}_{\{g_t\}}[\mathrm{median}(\{g_t\})|x_t]\|^2 - \|\nabla f(x_t) - \mathbb{E}_{\{g_t\}}[\mathrm{median}(\{g_t\})|x_t]\|^2\right)\right]$$
$$\quad + L\delta^2\mathbb{E}[\|\mathrm{median}(\{g_t\}) - \mathbb{E}[\mathrm{median}(\{g_t\})|x_t]\|^2 + \|\mathbb{E}[\mathrm{median}(\{g_t\})|x_t]\|^2]$$
$$= -\frac{\delta}{2}\mathbb{E}\left[\|\nabla f(x_t)\|^2 + \|\mathbb{E}[\mathrm{median}(\{g_t\})|x_t]\|^2 - \|\nabla f(x_t) - \mathbb{E}[\mathrm{median}(\{g_t\})|x_t]\|^2\right]$$
$$\quad + L\delta^2\mathbb{E}[\|\mathrm{median}(\{g_t\}) - \mathbb{E}[\mathrm{median}(\{g_t\})|x_t]\|^2 + \|\mathbb{E}[\mathrm{median}(\{g_t\})|x_t]\|^2]$$
$$= -\frac{\delta}{2}\mathbb{E}[\|\nabla f(x_t)\|^2] - (\frac{\delta}{2} - L\delta^2)\mathbb{E}[\|\mathbb{E}[\mathrm{median}(\{g_t\})|x_t]\|^2] + \frac{\delta}{2}\mathbb{E}[\|\nabla f(x_t) - \mathbb{E}[\mathrm{median}(\{g_t\})|x_t]\|^2]$$
$$\quad + L\delta^2\mathbb{E}[\|\mathrm{median}(\{g_t\}) - \mathbb{E}[\mathrm{median}(\{g_t\})|x_t]\|^2] \qquad (32)$$

where $\mathbb{E}_{x_t}[\cdot]$ is expectation over randomness of $x_t$ and $\mathbb{E}_{\{g_t\}}[\cdot|x_t]$ is expectation over randomness of $\{g_t\}$ given $x_t$.

Setting $\delta = \min(\frac{1}{\sqrt{Td}}, \frac{1}{2L})$, telescope sum and divide both sides by $T\delta/2$, we have

$$\frac{1}{T}\sum_{t=1}^{T}\mathbb{E}[\|\nabla f(x_t)\|^2]$$

$$\leq \frac{2\sqrt{d}}{\sqrt{T}}(\mathbb{E}[f(x_1)] - \mathbb{E}[f(x_{T+1})]) + \frac{1}{T}\sum_{t=1}^{T}\mathbb{E}[\|\nabla f(x_t) - \mathbb{E}[\text{median}(\{g_t\})|x_t]\|^2] + 2L\frac{\sqrt{d}}{\sqrt{T}}\sigma_m^2 \tag{33}$$

Substituting $\mathbb{E}[f(x_1)] - \mathbb{E}[f(x_{T+1})] \leq D_f$ into the above inequality completes the proof. $\qquad\square$

## D   Proof of Theorem 3

In this section, we show that our analysis is tight, in the sense that the constant gap

$$\frac{1}{T}\sum_{t=1}^{T}\mathbb{E}[\|\mathbb{E}[\text{median}(\{g_t\})]\| - \nabla f(x_t)\|_1] \tag{34}$$

does exist in practice.

Consider the following problem

$$\min_{x\in\mathbb{R}} f(x) \triangleq \frac{1}{3}\sum_{i=1}^{3}\frac{1}{2}(x - a_i)^2 \tag{35}$$

with $a_1 < a_2 < a_3$. In particular, $f_i(x) = \frac{1}{2}(x - a_i)^2$, so each local node has only one data point. Since the entire problem is deterministic, and the local gradient is also deterministic (i.e., no subsampling is available), we will drop the expectation below.

It is readily seen that the median of gradient is always $x - a_2$. Therefore running SIGNSGD on the above problem is equivalent to running SIGNSGD to minimize $\frac{1}{3}\sum_{i=1}^{3}\frac{1}{2}(x - a_2)^2$. From the Theorem 1 in Bernstein et al. [4], the SIGNSGD will converge to $x = a_2$ as $T$ goes to $\infty$ and $\delta = O(\frac{1}{\sqrt{T}}))$.

On the other hand, at the point $x = a_2$, the median of gradients $\text{median}(\{g_t\})$ is 0 but the gradient of $f(x)$ is given by

$$\nabla f(a_2) = \frac{1}{3}\sum_{i=1}^{3}(x - a_i) = \frac{1}{3}((a_2 - a_3) + (a_2 - a_1)) \tag{36}$$

Recall that for this problem, we also have for any $x_t$,

$$\|\mathbb{E}[\text{median}(\{g_t\})] - \nabla f(x_t)\|_1$$
$$= \|\text{median}(\{g_t\}) - \nabla f(x_t)\|_1$$
$$= \left|x_t - a_2 - \frac{1}{3}\sum_{i=1}^{3}(x_t - a_i)\right| = \left|\frac{1}{3}(2a_2 - a_1 - a_3)\right|. \tag{37}$$

Comparing (36) and (37), we conclude that at a given point $x = a_2$ (for which the SIGNSGD will converge to), we have

$$\|\nabla f(x)\|_1 = \frac{1}{T}\sum_{t=1}^{T}\mathbb{E}[\|\mathbb{E}[\text{median}(\{g_t\})]\| - \nabla f(x_t)\|_1] = \left|\frac{1}{3}(2a_2 - a_1 - a_3)\right|. \tag{38}$$

Substituting $a_1 = 0, a_2 = 1, a_3 = 5$ (which satisfies $a_1 < a_2 < a_3$ assumed the beginning) into (38) finishes the proof for SIGNSGD.

The proof for MEDIANSGD uses the same construction as the proof of Theorem 3, i.e. we consider the problem

$$\min_{x\in\mathbb{R}} f(x) \triangleq \frac{1}{3}\sum_{i=1}^{3}\frac{1}{2}(x - a_i)^2 \tag{39}$$

with $a_1 < a_2 < a_3$. Then from the update rule of MEDIANSGD, it reduces to running gradient descent to minimize $\frac{1}{2}(x - a_2)^2$. From classical results on convergence of gradient descent, the algorithm will converge to $x = a_2$ with any stepsize $\delta < 2/L$.

At the point $x = a_2$, the median of gradients is zero but $\nabla f(x)$ is

$$\nabla f(a_2) = \frac{1}{3}\sum_{i=1}^{3}(x - a_i) = \frac{1}{3}((a_2 - a_3) + (a_2 - a_1)). \tag{40}$$

In addition, for any $x_t$, the gap between median and mean of gradients satisfy

$$\|\mathbb{E}[\text{median}(\{g_t\})] - \nabla f(x_t)\|^2$$
$$= \left|x_t - a_2 - \frac{1}{3}\sum_{i=1}^{3}(x_t - a_i)\right|^2 = \left|\frac{1}{3}(2a_2 - a_1 - a_3)\right|^2 \tag{41}$$

Combining all above, we have for $x = a_2$, we get

$$\|\nabla f(x)\|^2 = \frac{1}{T}\sum_{t=1}^{T}\mathbb{E}[\|\mathbb{E}[\text{median}(\{g_t\})]] - \nabla f(x_t)\|^2] = \left|\frac{1}{3}(2a_2 - a_1 - a_3)\right|^2. \tag{42}$$

Setting $a_1 = 0, a_2 = 1, a_3 = 5$ we get $|\frac{1}{3}(2a_2 - a_1 - a_3)|^2 = 1$ and the proof is finished.

# E   Proof of Theorem 4

## E.1   Proof for (a)

Assume we have a set of numbers $u_1, .., u_{2n+1}$. Given a symmetric and unimodal noise distribution with mean 0 and variance 1, denote its pdf to be $h_0(z)$ and its cdf to be $H_0(z)$. Draw $2n + 1$ samples from the distribution $\xi_1, ..., \xi_{2n+1}$.

Given a constant $b$, define random variable $\hat{u}_i = u_i + b\xi_i$. Define $\tilde{u} \triangleq \text{median}(\{\hat{u}_i\}_{i=1}^{2n+1})$ and its pdf and cdf to be $h(z)$ and $H(z)$, respectively. Define $\bar{u} \triangleq \frac{1}{2n+1}\sum_{i=1}^{2n+1} u_i$.

Denote the pdf and cdf of $\hat{u}_i$ to be $h_i(z, b)$ and $H_i(z, b)$. Since $\hat{u}_i = u_i + b\xi_i$ is a scaled and shifted version of $\xi_i$, given $\xi_i$ has pdf $h_0(z)$ and cdf $H_0(z)$, we know $h_i(z, b) = \frac{1}{b}h_0(\frac{z-u_i}{b})$ and $H_i(z, b) = H_0(\frac{z-u_i}{b})$ from basic probability theory. In addition, from symmetricity of $h_0(z)$, we also have $1 - H_0(z) = H_0(-z)$.

Define pdf of $\tilde{u}$ to be $h(z, b)$, from order statistic, we know

$$h(z, b) = \sum_{i=1}^{2n+1} h_i(z, b) \sum_{S \in \mathcal{S}_i} \prod_{j \in S} H_j(z, b) \prod_{k \in [2n+1]\setminus\{i,S\}} (1 - H_k(z, b)) \tag{43}$$

where $\mathcal{S}_i$ is the set of all $n$-combinations of items from the set $[2n + 1] \setminus i$.

To simplify notation, we write the pdf into a more compact form

$$h(z, b) = \sum_{i,\{J,K\} \in \mathcal{S}_i'} h_i(z, b) \prod_{j \in J} H_j(z, b) \prod_{k \in K} (1 - H_k(z, b)) \tag{44}$$

where the set $\mathcal{S}_i'$ is the set of all possible $\{J, K\}$ with $J$ being a combination of $n$ items from $[2n + 1] \setminus i$ and $K = [2n + 1] \setminus \{J, i\}$ and $i \in [2n + 1]$ is omitted.

Then the expectation of median can be calculated as

$$
\mathbb{E}[\tilde{u}]
$$

$$
= \int_{-\infty}^{\infty} z \sum_{i,\{J,K\}\in\mathcal{S}_i'} h_i(z,b) \prod_{j\in J} H_j(z,b) \prod_{k\in K} (1 - H_k(z,b)) dz
$$

$$
= \sum_{i,\{J,K\}\in\mathcal{S}_i'} \int_{-\infty}^{+\infty} (bz + u_i)\frac{1}{b}h_0(z) \prod_{j\in J} H_0(z + \frac{u_i - u_j}{b}) \prod_{k\in K} (1 - H_0(z + \frac{u_i - u_k}{b}))bdz
$$

$$
= \sum_{i,\{J,K\}\in\mathcal{S}_i'} \int_{-\infty}^{+\infty} (bz + u_i)h_0(z)
$$

$$
\prod_{j\in J} \left( H_0(z) + \frac{u_i - u_j}{b}h_0(z) + \frac{(u_i - u_j)^2}{2b^2}h_0'(z_j') \right) \prod_{k\in K} \left( 1 - H_0(z) - \frac{u_i - u_k}{b}h_0(z) - \frac{(u_i - u_j)^2}{2b^2}h_0'(z_k') \right) dz
$$

where the second inequality is due to a changed of variable from $z$ to $\frac{z-u_i}{b}$, the last inequality is due to Taylor expansion and $z_j' \in [z_j, z_j + \frac{u_i-u_j}{\sigma}]$, $z_k' \in [z_k, z_k + \frac{u_i-u_k}{\sigma}]$.

Now we consider terms with different order w.r.t $b$ after expanding the Taylor expansion.

First, we start with the terms that is multiplied by $b$, the summation of coefficients in front of these terms equals

$$
\sum_{i,\{J,K\}\in\mathcal{S}_i'} \int_{-\infty}^{+\infty} zh_0(z) \prod_{j\in J} H_0(z)^n \prod_{k\in K} (1 - H_0(z))^n dz = 0
$$

due to symmetricity of $f$ over 0.

Then we consider the terms that are not multiplied by $b$, the summation of their coefficients equals

$$
\sum_{i,\{J,K\}\in\mathcal{S}_i'} (u_i - u_j)(\int_{-\infty}^{+\infty} zh_0(z) \prod_{j\in J} H_0(z)^{n-1} \prod_{k\in K} (1 - H_0(z))^n h_0(z)dz)
$$

$$
- \sum_{i,\{J,K\}\in\mathcal{S}_i'} (u_i - u_j)(\int_{-\infty}^{+\infty} zh_0(z) \prod_{j\in J} H_0(z)^n \prod_{k\in K} (1 - H_0(z))^{n-1} h_0(z)dz)
$$

$$
+ \sum_{i,\{J,K\}\in\mathcal{S}_i'} u_i(\int_{-\infty}^{+\infty} h_0(z)H_0(z)^n(1 - H_0(z))^n dz)
$$

$$
= 0 + 0 + \sum_{i=1}^{2n+1} u_i \binom{2n}{n} \int_{-\infty}^{+\infty} H_0(z)^n(1 - H_0(z))^n dH_0(z)
$$

due to the cancelling in the summation (i.e. $\sum_{i,\{J,K\}\in\mathcal{S}_i'} (u_i - u_j) = 0$).

Further, we have

$$\sum_{i=1}^{2n+1} u_i \binom{2n}{n} \int_{-\infty}^{+\infty} H_0(z)^n (1 - H_0(z))^n dH_0(z)$$

$$\overset{(a)}{=} \sum_{i=1}^{2n+1} u_i \binom{2n}{n} \int_0^1 y^n (1-y)^n dy$$

$$= \sum_{i=1}^{2n+1} u_i \binom{2n}{n} \frac{1}{n+1} \int_0^1 (1-y)^n dy^{n+1}$$

$$= \sum_{i=1}^{2n+1} u_i \binom{2n}{n} \frac{1}{n+1} \left( - \int_0^1 y^{n+1} d(1-y)^n \right)$$

$$\overset{(b)}{=} \sum_{i=1}^{2n+1} u_i \binom{2n}{n} \frac{n}{n+1} \int_0^1 y^{n+1} (1-y)^{n-1} dy$$

$$= \cdots$$

$$= \sum_{i=1}^{2n+1} u_i \binom{2n}{n} \frac{n(n-1)\cdots 1}{(n+1)(n+2)\cdots 2n} \int_0^1 y^{2n} dy$$

$$= \sum_{i=1}^{2n} u_i \binom{2n}{n} \frac{n!n!}{(2n+1)!} = \frac{1}{2n+1} \sum_{i=1}^{2n+1} u_i$$

where (a) is due to a change of variable from $H_0(z)$ to $y$ and the omitted steps are just repeating steps from $(a)$ to $(b)$.

In the last step, we consider the rest of the terms (terms multiplied by $1/b$ or higher order w.r.t. $1/b$). Since $h_0, h_0'$ are bounded, for any non-negative integer $p, q, k$, there exists a constant $c > 0$ such that:

$$\left| \int_{-\infty}^{+\infty} z h_0(z)(H_0(z)^p h_0(z)^q h_0'(z')^k) dz \right| \leq \int_{-\infty}^{+\infty} |z| |h_0(z)| |(H_0(z)^p h_0(z)^q h_0'(z')^k)| dz$$

$$\leq c \int_{-\infty}^{+\infty} |z| h_0(z) dz$$

$$= c \left( \int_{-1}^{+1} |z| h_0(z) dz + \int_1^{+\infty} |z| h_0(z) dz + \int_{-\infty}^{-1} |z| h_0(z) dz \right)$$

$$\leq c \left( \int_{-1}^{+1} h_0(z) dz + \int_1^{+\infty} z^2 h_0(z) dz + \int_{-\infty}^{-1} z^2 h_0(z) dz \right)$$

$$\leq c \left( \int_{-1}^{+1} h_0(z) dz + \int_{-\infty}^{+\infty} z^2 h_0(z) dz \right)$$

$$\leq c \left( \int_{-1}^{+1} h_0(z) dz + 1 \right)$$

$$\leq c \quad \text{[Here's another constant still denoted as } c\text{]}$$

And also

$$\left| \int_{-\infty}^{+\infty} h_0(z)(H_0(z)^p h_0(z)^q h_0'(z')^k) dz \right| \leq \int_{-\infty}^{+\infty} h_0(z) |H_0(z)^p h_0(z)^q h_0'(z')^k| dz \leq c' \int_{-\infty}^{+\infty} h_0(z) dz = c'$$

for some constant $c'$.

Then the coefficient of rest of the terms are bounded by constant, and the order of them are at least $\mathcal{O}(\frac{1}{b})$. Therefore $|\mathbb{E}[\text{median}(\{\hat{u}_i\}_{i=1}^{2n+1}) - \frac{1}{2n+1} \sum_{i=1}^{2n+1} u_i]| = \mathcal{O}(\frac{1}{b})$ which proves (6).

Now we compute the order of the variance of $\text{median}(\hat{u}_i)$ in terms of $b$

$\text{Var}(\text{median}(\hat{u}_i))$

$=\mathbb{E}[\text{median}(\hat{u}_i)^2] - \mathbb{E}[\text{median}(\hat{u}_i)]^2$

$\leq\mathbb{E}[\text{median}(\hat{u}_i)^2]$

$$= \sum_{i,\{J,K\}\in\mathcal{S}'_i}\int_{-\infty}^{+\infty} z^2 h_i(z)\prod_{j\in J}H_j(z)\prod_{k\in K}(1-H_k(z))dz$$

$$= \sum_{i,\{J,K\}\in\mathcal{S}_i}\int_{-\infty}^{+\infty} (bz+u_i)^2 h_0(z)\prod_{j\in J}H_0(z+\frac{u_i-u_j}{b})\prod_{k\in K}(1-H_0(z+\frac{u_i-u_k}{b}))dz$$

$$= \sum_{i,\{J,K\}\in\mathcal{S}_i}\int_{-\infty}^{+\infty} (bz+u_i)^2 h_0(z)\times$$

$$\prod_{j\in J}\left(H_0(z)+\frac{u_i-u_j}{b}h_0(z)+\frac{(u_i-u_j)^2}{2b^2}h'_0(z'_j)\right)\prod_{k\in K}\left(1-H_0(z)-\frac{u_i-u_k}{b}h_0(z)-\frac{(u_i-u_j)^2}{b^2}h'_0(z'_k)\right)dz$$

where $z'_j\in[z_j,z_j+\frac{u_i-u_j}{b}], z'_k\in[z_k,z_k+\frac{u_i-u_k}{b}]$. Similar to the analysis in computing order of gap between median and mean, we consider terms after expanding the multiple formula. Note that we similarly have:

$$\left|\int_{-\infty}^{+\infty} z^2 h_0(z)(H_0(z)^p h_0(z)^q h'_0(z')^k)dz\right| \leq \int_{-\infty}^{+\infty} z^2 h_0(z)|(H_0(z)^p h_0(z)^q h'_0(z')^k)|dz$$

$$\leq c\int_{-\infty}^{+\infty} z^2 h_0(z)dz$$

$$= c$$

Therefore, after expansion and integration, the coefficients of any order of $b$ are also bounded by constant. Since the order of the terms w.r.t $b$ are less than 2, we can conclude that the variance of $\text{Median}(\hat{u}_i)$ is of order $\mathcal{O}(b^2)$ which proves (7).

### E.2 Proof for (b)

This key idea of the proof in part is similar to that for part (a). We use Taylor expansion to expand different terms in pdf of sample median and identify the coefficient in front terms with different order w.r.t. $b$. The difference is that instead of doing second order Taylor expansion on $H_0$, we also need to do it for $h_0$, thus requiring $h''_0$ to be uniformly bounded and absolutely integrable. In addition, not every higher order term is multiplied by $h_0(z)$, thus more efforts are required for bounding the integration of higher order terms.

First, by a change of variable (change $z$ to $\frac{z-\bar{u}}{b}$), (43) can be written as

$$h(\bar{u}+bz,b) = \sum_{i=1}^{2n+1}\frac{1}{b}h_0(\frac{\bar{u}-u_i}{b}+z)\sum_{S\in\mathcal{S}_i}\prod_{j\in S}H_0(\frac{\bar{u}-u_j}{b}+z)\prod_{k\in[2n+1]\setminus\{i,S\}}H_0(-\frac{\bar{u}-u_k}{b}-z)$$

(45)

Using the Taylor expansion on $f$, we further have

$$h_0(\frac{\bar{u}-u_i}{b}+z) = h_0(z)+h'_0(z)(\frac{\bar{u}-u_i}{b})+\frac{h''_0(z_1)}{2}(\frac{\bar{u}-u_i}{b})^2$$

(46)

with $z_1\in(z,\frac{\bar{u}-u_i}{b}+z)$ or $z_1\in(\frac{\bar{u}-u_i}{b}+z,z)$. Similarly, we have

$$H_0(\frac{\bar{u}-u_j}{b}+z) = H_0(z)+h_0(z)(\frac{\bar{u}-u_j}{b})+\frac{h'_0(z_2)}{2}(\frac{\bar{u}-u_j}{b})^2$$

(47)

and

$$H_0(-\frac{\bar{u}-u_k}{b}-z) = H_0(-z)-h_0(-z)(\frac{u_k-\bar{u}}{b})-\frac{h'_0(-z_3)}{2}(\frac{u_k-\bar{u}}{b})^2$$

(48)

where $z_2 \in (z, \frac{\bar{u}-u_j}{b} + z)$ or $z_2 \in (\frac{\bar{u}-u_j}{b} + z, z)$, $z_3 \in (z, \frac{u_k-\bar{u}}{b} + z)$ or $z_3 \in (\frac{u_k-\bar{u}}{b} + z, z)$.

Substituting (46), (47), and (48) into (45), following similar argument as one can notice following facts.

1. Summation of all terms multiplied by $1/b$ is

$$\frac{1}{b} \sum_{i=1}^{2n+1} h_0(z) \sum_{S \in \mathcal{S}_i} \prod_{j \in S} H_0(z) \prod_{k \in [n] \setminus \{i, S\}} H_0(-z) \tag{49}$$

2. All the terms multiplied by $1/b^2$ cancels with each other after summation due to the definition of $\bar{u}$. I.e.

$$\sum_{i=1}^{2n+1} \frac{1}{b} h_0'(z)(\frac{\bar{u}-u_i}{b}) \sum_{S \in \mathcal{S}_i} H_0(z)^n H_0(-z)^n = 0 \tag{50}$$

$$\sum_{i=1}^{2n+1} \frac{1}{b} h_0(z) \sum_{S \in \mathcal{S}_i} \sum_{j \in S} h_0(z)(\frac{\bar{u}-u_j}{b}) H_0(z)^{n-1} H_0(-z)^n = 0 \tag{51}$$

$$\sum_{i=1}^{2n+1} \frac{1}{b} h_0(z) \sum_{S \in \mathcal{S}_i} H_0(z)^n \sum_{k \in [n] \setminus \{i, S\}} h_0(-z)(\frac{\bar{u}-u_k}{b}) H_0(-z)^{n-1} = 0 \tag{52}$$

3. Excluding the terms above, the rest of the terms are upper bounded by the order of $O(1/b^3)$.

Thus by another change of variable (change $z$ to $\frac{z}{b}$), we have

$$h(\bar{u} + z, b) = \frac{1}{b} g(\frac{z}{b}) + \frac{1}{b} v(\frac{z}{b}) \tag{53}$$

where

$$g(z) = \sum_{i=1}^{2n+1} h_0(z) \sum_{S \in \mathcal{S}_i} \prod_{j \in S} H_0(z) \prod_{k \in [n] \setminus \{i, S\}} H_0(-z) \tag{54}$$

which is the pdf of sample median of $2n + 1$ iid draws from $h_0$ and it is symmetric around $0$.

Further, observe that when $h_0(z)$, $h_0'(z)$, and $h''(z)$ are all absolutely upper bounded and absolutely integrable, integration of absolute value of each high order term in $v(z)$ can be upper bounded in the order of $O(\frac{\max_i |\bar{u}-u_i|^2}{b^2})$. This is because each term in $v(z)$ is at least multiplied by $1/b^2$ and one of $h_0(z)$, $h_0(z)$, $h_0'(z)$, $h_0''(z_1)$, $h_0'(z_2)$ and $h_0'(-z_3)$ ($z_1$, $z_2$, $z_3$ appears through remainder terms of the Taylor' theorem). The terms multiplied by $h_0(z)$, $h_0(-z)$, or $h_0'(z)$ absolutely integrates into a constant. The terms multiplied only by the remainder terms in the integration are more tricky, one need to rewrite the remainder term into integral form and exchange the order of integration to prove that the term integrates the order of $O(1/b^2)$. We do this process for one term in the following and the others are omitted.

$$\int_{-\infty}^{\infty} \frac{h_0''(z_1)}{2} (\frac{\bar{u}-u_i}{b})^2 H_0(z) H_0(-z) dx$$

$$\leq \int_{-\infty}^{\infty} \left| \frac{h_0''(z_1)}{2} (\frac{\bar{u}-u_i}{b})^2 \right| \|H_0\|_\infty \|H_0\|_\infty dx$$

$$= \|H_0\|_\infty \|H_0\|_\infty \int_{-\infty}^{\infty} \left| \int_x^{x+\frac{\bar{u}-u_i}{b}} h_0''(t)(t-x) dt \right| dx \tag{55}$$

where the equality holds because $\frac{h_0''(z_1)}{2} (\frac{\bar{u}-u_i}{b})^2$ is the remainder term of the Taylor expansion when approximating $z + \frac{\bar{u}-u_i}{b}$ at $z$ and we changed the remainder term from the mean-value form to the integral form.

Without loss of generality, we assume $\bar{u} - u_i \geq 0$ (the proof is similar when it is less than 0), then we get

$$\int_{-\infty}^{\infty} \left| \int_x^{x+\frac{\bar{u}-u_i}{b}} h_0''(t)(t-x)dt \right| dx$$

$$\leq \int_{-\infty}^{\infty} \int_x^{x+\frac{\bar{u}-u_i}{b}} |h_0''(t)||(t-x)|dtdx$$

$$= \int_{-\infty}^{\infty} \int_{t-\frac{\bar{u}-u_i}{b}}^t |h_0''(t)||(t-x)|dxdt$$

$$= \frac{1}{2}(\frac{\bar{u}-u_i}{b})^2 \int_{-\infty}^{\infty} |h_0''(t)| \, dt \qquad (56)$$

which is $(\frac{\bar{u}-u_i}{b})^2$ times a constant.

Thus, we have

$$\int_{-\infty}^{\infty} \frac{1}{b}|v(\frac{z}{b})|dz = \int_{-\infty}^{\infty} \frac{1}{b}|v(z)|bdz = O(\frac{\max_i |\bar{u}-u_i|^2}{b^2}) \qquad (57)$$

which completes this part. $\qquad\qquad\qquad\qquad\qquad\qquad\qquad\qquad\qquad\qquad\qquad\square$

## F   An extended version of of Theorem 4

Before presenting the proof of convergence of SIGNSGD and MEDIANSGD, we need an version of Theorem 4 with stochastic sampling, we present the theorem and its proof in this section.

**Theorem F.1.** *Assume we have 2n+1 set of numbers $A_1 = \{a_{1,j}\}_{j=1}^{k_1}, A_2 = \{a_{2,j}\}_{j=1}^{k_1}, ... A_{2n+1} = \{a_{2n+1,j}\}_{j=1}^{k_{2n+1}}$ with mean of the numbers of each set being $u_1, .., u_{2n+1}$. Given a symmetric and unimodal noise distribution with mean 0, variance 1. Denote the pdf of the distribution to be $h_0(z)$ and cdf to be $H_0(z)$. Suppose $h_0'(z)$ is uniformly bounded and absolutely integrable. Draw $2n+1$ samples $\xi_1, ..., \xi_{2n+1}$ from the distribution $h_0(z)$. Define random variable $q_i$ to be a number uniformly randomly drawn from $\{a_{i,j}\}_{j=1}^{K_i}$ and $\hat{q}_i = q_i + b\xi_i$, $\bar{u} \triangleq \sum_{i=1}^{2n+1} u_i$,*

*(a) We have*

$$\mathbb{E}[\mathrm{median}(\{\hat{q}_i\}_{i=1}^{2n+1})] = \bar{u} + O\left(\frac{\max_{i,j,i',j'|i\neq i'} |a_{i,j}-a_{i',j'}|^2}{b}\right), \qquad (58)$$

$$\mathrm{Var}(\mathrm{median}(\{\hat{q}_i\}_{i=1}^{2n+1})) = O(b^2). \qquad (59)$$

*(b) Further assume $h_0''(z)$ is uniformly bounded and absolutely integrable. Denote $r_b(z)$ to be the pdf of the distribution of $\mathrm{median}(\{\hat{q}_i\}_{i=1}^{2n+1})$ and $S_A = A_1 \times A_2 \times ... \times A_{2n+1}$, we have*

$$r_b(\bar{u}+z) = \underbrace{\frac{1}{b}g(\frac{z}{b})}_{\text{symmetric part}} + \underbrace{\frac{1}{b}v(\frac{z}{b})}_{\text{asymmetric part}} \qquad (60)$$

*where*

$$g(z) \triangleq \sum_{i=1}^{2n+1} h_0(z) \sum_{S\in\mathcal{S}_i} \prod_{j\in S} H_0(z) \prod_{k\in[n]\setminus\{i,S\}} H_0(-z) \qquad (61)$$

*being the pdf of sample median of $2n+1$ samples drawn from the distribution $h_0(z)$ which is symmetric over 0, $\mathcal{S}_i$ is the set of all $n$-combinations of items from the set $[2n+1] \setminus i$, and the asymmetric part satisfies*

$$\int_{-\infty}^{\infty} \frac{1}{b}|v(\frac{z}{b})|dz = O\left(\frac{\mathbb{E}_{s\sim U(S_A)}[\max_i |s_i-\bar{s}|^2]}{b^2}\right) + O\left(\frac{\max_{s\in S_A}(\bar{s}-\bar{u})^2}{b^2}\right) \qquad (62)$$

*where $U(S_A)$ is uniform distribution over elements in $S_A$*

**Proof of Theorem F.1**: The proof is mostly based on Theorem 4 with some extra efforts dealing with the sampling noise. We first prove part (a) (58). Since $q_i$ is sampled uniformly randomly from $\{a_{i,j}\}_{j=1}^{K_i}$, we know there are $\prod_{i=1}^{2n+1} K_i$ possible realizations for $\{q_i\}_{i=1}^{2n+1}$ with equal probability. I addition, the mean of mean of these realizations is $\bar{u}$. For each realization $\{q_i = \tilde{q}_i\}_{i=1}^{2n+1}$, we know from Theorem 4 (a) that

$$\mathbb{E}[\text{median}(\{\hat{q}_i\}_{i=1}^{2n+1})|\{q_i = \tilde{q}_i\}_{i=1}^{2n+1}] = \frac{1}{2n+1}\sum_{i=1}^{2n+1} \tilde{q}_i + O\left(\frac{\max_{i,i'}|\tilde{q}_i - \tilde{q}_{i'}|^2}{b}\right). \quad (63)$$

Given the fact that $\mathbb{E}[\text{median}(\{\hat{q}_i\}_{i=1}^{2n+1})] = \mathbb{E}_{\{q_i\}_{i=1}^{2n+1}}[\mathbb{E}_{\{\xi_i\}_{j=1}^{2n+1}}[\text{median}(\{\hat{q}_i\}_{i=1}^{2n+1})|\{q_i\}_{i=1}^{2n+1}]]$ and $\mathbb{E}[\frac{1}{2n+1}\sum_{i=1}^{2n+1} q_i] = \bar{u}$, we know

$$\mathbb{E}[\text{median}(\{\hat{q}_i\}_{i=1}^{2n+1})] = \bar{u} + O\left(\frac{\max_{i,i',j,j',i\neq i'}|a_{i,i} - a_{i',j'}|^2}{b}\right). \quad (64)$$

 which proves (58).

Now we prove (59). We have

$$\text{Var}(\text{median}(\{\hat{q}_i\}_{i=1}^{2n+1}))$$
$$=\mathbb{E}[(\text{median}(\{\hat{q}_i\}_{i=1}^{2n+1}) - \mathbb{E}[\text{median}(\{\hat{q}_i\}_{i=1}^{2n+1})])^2]$$
$$=\mathbb{E}_{\{q_i\}_{j=1}^{2n+1}}[\mathbb{E}_{\{\xi_i\}_{j=1}^{2n+1}}[(\text{median}(\{\hat{q}_i\}_{i=1}^{2n+1}) - \mathbb{E}[\text{median}(\{\hat{q}_i\}_{i=1}^{2n+1})])^2|\{q_i\}_{j=1}^{2n+1}]]$$
$$\leq\mathbb{E}_{\{q_i\}_{j=1}^{2n+1}}[\mathbb{E}_{\{\xi_i\}_{j=1}^{2n+1}}[2(\text{median}(\{\hat{q}_i\}_{i=1}^{2n+1}) - \mathbb{E}_{\{\xi_i\}_{j=1}^{2n+1}}[\text{median}(\{\hat{q}_i\}_{i=1}^{2n+1})|\{q_i\}_{j=1}^{2n+1}])^2|\{q_i\}_{j=1}^{2n+1}]]$$
$$+ \mathbb{E}_{\{q_i\}_{j=1}^{2n+1}}[\mathbb{E}_{\{\xi_i\}_{j=1}^{2n+1}}[2(\mathbb{E}_{\{\xi_i\}_{j=1}^{2n+1}}[\text{median}(\{\hat{q}_i\}_{i=1}^{2n+1})|\{q_i\}_{j=1}^{2n+1}] - \mathbb{E}[\text{median}(\{\hat{q}_i\}_{i=1}^{2n+1})])^2|\{q_i\}_{j=1}^{2n+1}]]$$
$$=2\mathbb{E}_{\{q_i\}_{j=1}^{2n+1}}[\text{Var}(\text{median}(\{\hat{q}_i\}_{i=1}^{2n+1})|\{q_i\}_{j=1}^{2n+1})]$$
$$+ 2\mathbb{E}_{\{q_i\}_{j=1}^{2n+1}}[(\mathbb{E}_{\{\xi_i\}_{j=1}^{2n+1}}[\text{median}(\{\hat{q}_i\}_{i=1}^{2n+1})|\{q_i\}_{j=1}^{2n+1}] - \mathbb{E}[\text{median}(\{\hat{q}_i\}_{i=1}^{2n+1})])^2]$$
$$\leq O(b^2) + O(1) \quad (65)$$

where the last inequality is because we can apply (7) to the variance term and for the remaining term we have (63) and (64).

Now we prove part (b) of the theorem. Denote $A_i = \{a_i\}_{i=1}^{K_i}$ and $S_A = A_1 \times A_2 \times ... \times A_{2n+1}$. Also, given a vector $s \in R^{2n+1}$, denote $r_b(z, s)$ to be the pdf of $\text{median}(\{\hat{q}_i\}_{i=1}^{2n+1})$ with $q_i = s_i, \forall i \in [2n+1]$. By definition of $\text{median}(\{\hat{q}_i\}_{i=1}^{2n+1})$, we know know

$$r_b(z) = \frac{1}{|S_A|}\sum_{s \in S_A} r_b(z, s). \quad (66)$$

Denote $\bar{s} = \frac{1}{2n+1}\sum_{i=1}^{2n+1} s_i$, from Theorem 4 (b), we know

$$r_b(\bar{s} + z, s) = \underbrace{\frac{1}{b}g(\frac{z}{b})}_{\text{symmetric part}} + \underbrace{\frac{1}{b}v(\frac{z}{b})}_{\text{asymmetric part}}$$

with

$$g(z) = \sum_{i=1}^{2n+1} h_0(z)\sum_{S \in \mathcal{S}_i}\prod_{j \in S} H_0(z)\prod_{k \in [n]\setminus\{i,S\}} H_0(-z)$$

being the pdf of sample median of $2n + 1$ samples drawn from the distribution $h_0(z)$ which is symmetric over 0, $\mathcal{S}_i$ is the set of all $n$-combinations of items from the set $[2n + 1] \setminus i$, and

$$\int_{-\infty}^{\infty}\frac{1}{b}|v(\frac{z}{b})|dz = O\left(\frac{\max_i|\bar{s} - s_i|^2}{b^2}\right)$$

By Taylor expansion, we know

$$r_b(\bar{u} + z, s) = \frac{1}{b}g(\frac{z + \bar{s} - \bar{u}}{b}) + \frac{1}{b}v(\frac{z + \bar{s} - \bar{u}}{b})$$

$$= \frac{1}{b}\left(g(\frac{z}{b}) + g'(\frac{z}{b})(\frac{\bar{s} - \bar{u}}{b}) + \frac{1}{2}g''(\frac{z_1}{b})(\frac{\bar{s} - \bar{u}}{b})^2\right) + \frac{1}{b}v(\frac{z + \bar{s} - \bar{u}}{b}) \quad (67)$$

with $z_1 \in [\min(z, z + \bar{s} - \bar{u}), \max(z, z + \bar{s} - \bar{u})]$

Substituting (67) into (66), we know

$$r_b(\bar{u} + z)$$

$$= \frac{1}{|S_A|}\sum_{s \in S_A} r_b(\bar{u} + z, s)$$

$$= \frac{1}{|S_A|}\sum_{s \in S_A}\frac{1}{b}\left(g(\frac{z}{b}) + g'(\frac{z}{b})(\frac{\bar{s} - \bar{u}}{b}) + \frac{1}{2}g''(\frac{z_1}{b})(\frac{\bar{s} - \bar{u}}{b})^2\right) + \frac{1}{|S_A|}\sum_{s \in S_A}\frac{1}{b}v(\frac{z + \bar{s} - \bar{u}}{b})$$

$$= \frac{1}{b}g(\frac{z}{b}) + \frac{1}{|S_A|}\sum_{s \in S_A}\frac{1}{b}\left(\frac{1}{2}g''(\frac{z_1}{b})(\frac{\bar{s} - \bar{u}}{b})^2\right) + \frac{1}{|S_A|}\sum_{s \in S_A}\frac{1}{b}v(\frac{z + \bar{s} - \bar{u}}{b}) \quad (68)$$

where we have used the fact that $\frac{1}{|S_A|}\sum_{s \in S_A}\bar{s} = \bar{u}$ in the last equality.

What remains is to bound the integration of terms other than $\frac{1}{b}g(\frac{z}{b})$ in (68).

From Theorem 4 (b), we already know

$$\int_{-\infty}^{\infty}\frac{1}{|S_A|}\sum_{s \in S_A}\left|\frac{1}{b}v(\frac{z + \bar{s} - \bar{u}}{b})\right|dz$$

$$\leq \frac{1}{|S_A|}\sum_{s \in S_A}O\left(\frac{\max_i |s_i - \bar{s}|^2}{b^2}\right)$$

$$= O\left(\frac{1}{|S_A|}\sum_{s \in S_A}\frac{\max_i |s_i - \bar{s}|^2}{b^2}\right) \quad (69)$$

Thus, we only need to bound $\frac{1}{|S_A|}\frac{1}{2b}\sum_{s \in S_A}\int_{-\infty}^{\infty}\left|g''(\frac{z_1}{b})(\frac{\bar{s}-\bar{u}}{b})^2\right|dz$. Each term in the summation can be upper bounded as $(\frac{(\bar{s}-\bar{u})^2}{b^2}\int_{-\infty}^{\infty}|g''(z)|d_z$ using the same procedure in (55) and (56). Thus, we have

$$\frac{1}{|S_A|}\sum_{s \in S_A}\int_{-\infty}^{\infty}\frac{1}{b}\left(\frac{1}{2}g''(\frac{z_1}{b})(\frac{\bar{s} - \bar{u}}{b})^2\right)dz = \frac{1}{|S_A|}\sum_{s \in S_A}O\left(\frac{(\bar{s} - \bar{u})^2}{b^2}\right) = O\left(\frac{\mathbb{E}_{s \in S_A}(\bar{s} - \bar{u})^2}{b^2}\right)$$
$$\quad (70)$$

which proves the theorem. $\qquad\square$

# G  Proof of Theorem 5

Now we are ready to prove Theorem 5.

Use the fact that the noise on median is approximately unimodal and symmetric, one may prove that SIGNSGD can converge to a stationary point. With symmetric and unimodal noise, the bias in SIGNSGD can be alternatively viewed as a decrease of effective learning rate, thus slowing down the optimization instead of leading a constant bias. This proof formalizes this idea by characterizing the asymmetricity of the noise ($O(1/\sigma^2)$) and then follows a sharp analysis for SIGNSGD. The key difference from Theorem 1 is taking care of the bias introduced by the difference between median and mean.

Let us recall:

$$\text{median}(\{g_t\}) \triangleq \text{median}(\{(g_{t,i}\}_{i=1}^M), \quad (71)$$

$$\text{median}(\{\nabla f_t\}) \triangleq \text{median}(\{\nabla f_i(x_t)\}_{i=1}^M). \quad (72)$$

where

$$g_{t,i} = \nabla f_i(x_t) + b\xi_{t,i} + \zeta_{t,i} \tag{73}$$

where $\xi_{t,i}$ is a $d$ dimensional random vector with each element drawn iid from $N(0,1)$ and we abuse the notation $\zeta_{t,i}$ to denote an zero-mean additive discrete noise caused by sampling on data.

By (23), we have the following series of inequalities

$$f(x_{t+1}) - f(x_t)$$

$$\leq -\delta\langle\nabla f(x_t), \text{sign}(\text{median}(\{g_t\}))\rangle + \frac{L}{2}\delta^2 d$$

$$= -\delta\sum_{j=1}^{d}|\nabla f(x_t)_j|(I[\text{sign}(\text{median}(\{g_t\})_j) = \text{sign}(\nabla f(x_t)_j)] - I[\text{sign}(\text{median}(\{g_t\})_j) \neq \text{sign}(\nabla f(x_t)_j)])$$

$$+ \frac{L}{2}\delta^2 d \tag{74}$$

where $\text{median}(\{g_t\})_j$ is $j$th coodrinate of $\text{median}(\{g_t\})$, and $I[\cdot]$ denotes the indicator function.

Taking expectation over all the randomness, we get

$$\mathbb{E}[f(x_{t+1})] - \mathbb{E}[f(x_t)]$$

$$\leq -\delta\mathbb{E}\left[\sum_{j=1}^{d}|\nabla f(x_t)_j|\left(P[\text{sign}(\text{median}(\{g_t\})_j) = \text{sign}(\nabla f(x_t)_j)] - P[\text{sign}(\text{median}(\{g_t\})_j) \neq \text{sign}(\nabla f(x_t)_j)]\right)\right]$$

$$+ \frac{L}{2}\delta^2 d \tag{75}$$

Now we need a refined analysis on the error probability. In specific, we need an sharp analysis on the following quantity

$$P[\text{sign}(\text{median}(\{g_t\})_j) = \text{sign}(\nabla f(x_t)_j)] - P[\text{sign}(\text{median}(\{g_t\})_j) \neq \text{sign}(\nabla f(x_t)_j)]. \tag{76}$$

Using reparameterization, we can rewrite $\text{median}(\{g_t\})$ as

$$\text{median}(\{g_t\}) = \nabla f(x_t) + \xi_t \tag{77}$$

where $\xi_t$ is created by $\xi_{t,i}$'s and $\zeta_{t,i}$'s added on the local gradients on different nodes.

Then, w.l.o.g., assume $\nabla f(x_t)_j \geq 0$ we have

$$P[\text{sign}(\text{median}(\{g_t\})_j) \neq \nabla f(x_t)_j]$$
$$= P[(\xi_t)_j \leq -\nabla f(x_t)_j]$$
$$= \int_{-\infty}^{-\nabla f(x_t)_j} h_{t,j}(z) \tag{78}$$

where $h_{t,j}(z)$ is the pdf of the $j$th coordinate of $\xi_t$.

Similarly, we have

$$P[\text{sign}(\text{median}(\{g_t\})_j) = \nabla f(x_t)_j]$$
$$= P[(\xi_t)_j > -\nabla f(x_t)_j]$$
$$= \int_{-\nabla f(x_t)_j}^{\infty} h_{t,j}(z) \tag{79}$$

From (9) and (10), we can split $h_{t,j}(z)$ into a symmetric part and a non-symmetric part which can be written as

$$h_{t,j}(z) = h_{t,j}^s(z) + h_{t,j}^u(z) \tag{80}$$

where $h_{t,j}^s(z)$ is symmetric around 0 and $h_{t,j}^u(z)$ is not.

Therefore, from (79) and (78), we know that

$$P[\text{sign}(\text{median}(\{g_t\})_j) = \text{sign}(\nabla f(x_t)_j)] - P[\text{sign}(\text{median}(\{g_t\})_j) \neq \text{sign}(\nabla f(x_t)_j)]$$

$$= \int_{-\nabla f(x_t)_j}^{\infty} h_{t,j}(z) - \int_{-\infty}^{-\nabla f(x_t)_j} h_{t,j}(z)$$

$$= \int_{-\nabla f(x_t)_j}^{\infty} h_{t,j}^s(z) + h_{t,j}^u(z) - \int_{-\infty}^{-\nabla f(x_t)_j} h_{t,j}^s(z) + h_{t,j}^u(z)$$

$$= \int_{-\nabla f(x_t)_j}^{\nabla f(x_t)_j} h_{t,j}^s(z) + \int_{-\nabla f(x_t)_j}^{\infty} h_{t,j}^u(z) - \int_{-\infty}^{-\nabla f(x_t)_j} h_{t,j}^u(z) \tag{81}$$

where the last equality is due to symmetricity of $h_{t,j}^s(z)$, and the assumption that $\nabla f(x_t)_j$ is positive. To simplify the notations, define a new variable $z_{t,j}$ with pdf $h_{t,j}^s$, then we have

$$\int_{-\nabla f(x_t)_j}^{\nabla f(x_t)_j} h_{t,j}^s(z) = P[|z_{t,j}| \leq |\nabla f(x_t)_j|] \tag{82}$$

A similar result can be derived for $\nabla f(x_t)_j \leq 0$.

In addition, since the noise on each coordinate of local gradient satisfy Theorem 4, we can apply (10) to each coordinate of the stochastic gradient vectors. Denote $\bar{\hat{g}}_t = \frac{1}{2n+1}\sum_{i=1}^{2n+1}\hat{g}_{t,i}$ we know that

$$\int_{-\infty}^{\infty} |h_{t,j}^u(z)| = O\left(\frac{\mathbb{E}_{\{\hat{g}_{t,i}\}_{i=1}^{2n+1}}[\max_i|(\hat{g}_{t,i})_j - (\bar{\hat{g}}_t)_j|^2]}{b^2}\right) + O\left(\frac{\mathbb{E}_{\{\hat{g}_{t,i}\}_{i=1}^{2n+1}}((\bar{\hat{g}}_t)_j - \nabla f(x_t)_j)^2}{b^2}\right) \tag{83}$$

and thus

$$\int_{-\nabla f(x_t)_j}^{\infty} h_{t,j}^u(z) - \int_{-\infty}^{-\nabla f(x_t)_j} h_{t,j}^u(z)$$

$$\leq O\left(\frac{\mathbb{E}_{\{\hat{g}_{t,i}\}_{i=1}^{2n+1}}[\max_i|(\hat{g}_{t,i})_j - (\bar{\hat{g}}_t)_j|^2]}{b^2}\right) + O\left(\frac{\mathbb{E}_{\{\hat{g}_{t,i}\}_{i=1}^{2n+1}}((\bar{\hat{g}}_t)_j - \nabla f(x_t)_j)^2}{b^2}\right)$$

$$= O(\frac{1}{b^2}). \tag{84}$$

**Remark:** We comment that (84) can be small if a large minibatch is used and $\max_i \|\nabla f_i(x_t) - \nabla f(x_t)\|$ is small even with a constant $b$. Consider when full batch gradient evaluation is used. In this case, we know $\hat{g}_{t,i} = \nabla f_i(x_t)$ and the second term on RHS of the first inequality of (84) become 0. The first term becomes $O\left(\frac{\max_i|\nabla f_i(x_t)_j - \nabla f(x_t)_j|^2}{b^2}\right)$ which can be bounded by $\max_i \|\nabla f_i(x_t) - \nabla f(x_t)\|$. With homogeneous data distribution one usually have $\max_i \|\nabla f_i(x_t) - \nabla f(x_t)\|$ being a small number. Getting back to the minibatch case where the stochastic gradients are evaluated on a minibatches, when the number of samples in a batch is large, we have $\hat{g}_t$ being close to its mean $\nabla f(x_t)$ with high probability by the law of large numbers which means the second term on RHS of (84) is small. Similarly, we have $\hat{g}_{t,i}$ approaching $\nabla f_i(x_t)$ and the first term becomes similar to the full batch case, which be small on a heterogeneous data distribution.

To continue, we need to introduce some new definitions. Define

$$W_{t,j} = \frac{|\nabla f(x_t)_j|}{b\sigma_{mid}} \tag{85}$$

where $\sigma_{mid}$ is the variance of the noise with pdf (9), that is

$$g(z) = \sum_{i=1}^{2n+1} h_0(z) \sum_{S \in \mathcal{S}_i} \prod_{j \in S} H_0(z) \prod_{k \in [n]\setminus\{i,S\}} H_0(-z) \tag{86}$$

By adapting Lemma 1 from Bernstein et al. [5] (which is an application of Gauss's inequality), we have

$$P[|z_{t,j}| < |\nabla f(x_t)_j|] \geq \begin{cases} 1/3 & W_{t,j} \geq \frac{2}{\sqrt{3}} \\ \frac{W_{t,j}}{\sqrt{3}} & \text{otherwise} \end{cases} \quad (87)$$

Thus, continuing from (75), we have

$$\mathbb{E}[f(x_{t+1})] - \mathbb{E}[f(x_t)]$$

$$\leq -\delta\mathbb{E}\left[\sum_{j=1}^{d}|\nabla f(x_t)_j|(P[\text{sign}(\text{median}(\{g_t\})_j) = \text{sign}(\nabla f(x_t)_j)] - P[\text{sign}(\text{median}(\{g_t\})_j) \neq \text{sign}(\nabla f(x_t)_j)])\right]$$

$$+ \frac{L}{2}\delta^2 d$$

$$\leq -\delta\mathbb{E}\left[\sum_{j=1}^{d}|\nabla f(x_t)_j|(P[|z_{t,j}| < |\nabla f(x_t)_j|])\right]$$

$$+ \delta\mathbb{E}\left[\sum_{j=1}^{d}|\nabla f(x_t)_j|O\left(\frac{1}{b^2}\right)\right] + \frac{L}{2}\delta^2 d \quad (88)$$

Define $D_f \triangleq f(x_1) - \min_x f(x)$, telescope from 1 to $T$, divide both sides by $T\delta$, we have

$$\frac{1}{T}\sum_{t=1}^{T}\mathbb{E}\left[\sum_{j=1}^{d}|\nabla f(x_t)_j|(P[|z_{t,j}| < |\nabla f(x_t)_j|])\right]$$

$$\leq \frac{1}{T\delta}D_f + \frac{1}{T}\sum_{t=1}^{T}\mathbb{E}\left[\sum_{j=1}^{d}|\nabla f(x_t)_j|O\left(\frac{1}{b^2}\right)\right] + \frac{L}{2}\delta d \quad (89)$$

where the RHS is decaying with a speed of $\frac{\sqrt{d}}{\sqrt{T}}$.

Further, substituting (87) and (85) into (89) and multiplying both sides of (89) by 3, we can get

$$\frac{1}{T}\sum_{t=1}^{T}\left(\sum_{j\in\mathcal{W}_t}|\nabla f(x_t)_j| + \frac{1}{b\sigma_{mid}}\sum_{j\in[d]\backslash\mathcal{W}_t}\nabla f(x_t)_j^2\right)$$

$$\leq 3\frac{1}{T\delta}D_f + 3\frac{1}{T}\sum_{t=1}^{T}\mathbb{E}\left[\sum_{j=1}^{d}|\nabla f(x_t)_j|O\left(\frac{1}{b^2}\right)\right] + 3\frac{L}{2}\delta d \quad (90)$$

which completes the proof. $\qquad\square$

## H  Proof of Theorem 6

Following the same procedures as the Theorem 2, we can get

$$\mathbb{E}[f(x_{t+1})] - \mathbb{E}[f(x_t)]$$

$$\leq -\frac{\delta}{2}\mathbb{E}[\|\nabla f(x_t)\|^2] - (\frac{\delta}{2} - L\delta^2)\mathbb{E}[\|\mathbb{E}[\text{median}(\{g_t\})|x_t]\|^2] + \frac{\delta}{2}\mathbb{E}[\|\nabla f(x_t) - \mathbb{E}[\text{median}(\{g_t\})|x_t]\|^2]$$

$$+ L\delta^2\mathbb{E}[\|\text{median}(\{g_t\}) - \mathbb{E}[\text{median}(\{g_t\})|x_t]\|^2] \quad (91)$$

which is the same as (32).

Sum over $t \in [T]$ and divide both sides by $T\delta/2$, assume $\delta \leq \frac{1}{2L}$, we get

$$\frac{1}{T}\sum_{t=1}^{T}\mathbb{E}[\|\nabla f(x_t)\|^2]$$

$$\leq \frac{2}{T\delta}(\mathbb{E}[f(x_1)] - \mathbb{E}[f(x_{T+1})]) + +\frac{1}{T}\sum_{t=1}^{T}\mathbb{E}[\|\nabla f(x_t) - \mathbb{E}[\text{median}(\{g_t\})|x_t]\|^2] + 2Ld\delta\sigma_m^2$$

$$(92)$$

By (58) in Theorem F.1, we know

$$\mathbb{E}[\|\nabla f(x_t) - \mathbb{E}[\text{median}(\{g_t\})|x_t]\|] = O(\frac{\sqrt{d}}{b}) \tag{93}$$

where $\sqrt{d}$ is due to $L_2$ norm. In addition, we have $\sigma_m^2 = O(b^2)$ by (59) in Theorem F.1. Assume $\mathbb{E}[\text{median}(\{g_t\})_j|x_t] \leq Q$ and set $b = T^{1/6}d^{1/6}$ and $\delta = T^{-2/3}d^{-2/3}$, we get

$$\frac{1}{T}\sum_{t=1}^{T}\mathbb{E}[\|\nabla f(x_t)\|^2]$$

$$\leq \frac{2}{T\delta}(\mathbb{E}[f(x_1)] - \mathbb{E}[f(x_{T+1})]) + O\left(\frac{d}{b^2}\right) + O\left(\delta d b^2\right). \tag{94}$$

Then, upper bounding $\mathbb{E}[f(x_1)] - \mathbb{E}[f(x_{T+1})])$ by $D_f$ finishes the proof. $\qquad\square$

# I   Details of the Implementation

Our experimentation is mainly implemented using Python 3.6.4 with packages MPI4Py 3.0.0, NumPy 1.14.2 and TensorFlow 1.10.0. We use the Message Passing Interface (MPI) to implement the distributed system, and use TensorFlow to implement the neural network. The MNIST experiments are run on up to 20 compute cores of two Intel Haswell E5-2680 CPUs with 64 GB Memory. The CIFAR-10 experiments are run on up to 5 AWS p3.2xlarge machines.

## I.1   Dataset and pre-processing

In the first experiment, we use the MNIST dataset[2], which contains a training set of 60,000 samples, and a test set of 10,000 samples, both are 28x28 grayscale images of the 10 handwritten digits. To facilitate the neural network training, the original feature vector, which contains the integer pixel value from 0 to 255, has been scaled to a float vector in the range $(0,1)$. The integer categorical label is also converted to the binary class matrix (one hot encoding) for use with the categorical cross-entropy loss. For the CIFAR-10 dataset, the data are processed in the same way.

## I.2   Neural Network and Initialization

For MNIST, a two-layer fully connected neural network with 128 and 10 neurons for each layer is used in the experiment. The initialization parameters are drawn from a truncated normal distribution centered on zero, with variance scaled with the number of input units in the weight tensor (fan-in). For CIFAR-10, we use ResNet-20 (obtained from the implementation of ResNet20 v1 in Keras) with batch normalization layers removed, parameters are initialized the same way as in Keras. We removed the batch normalization layers because inconsistency of statistics (mean and variance) of batch normalization layers on different nodes significantly deteriorates the performance when heterogeneous data is used. This phenomenon is also observed and explained in [11, 12].

## I.3   Parameter Tuning

We use constant stepsize for MNIST and a learning rate schedule for CIFAR-10. For MNIST, the stepsize is chosen from the set $\{1, 0.1, 0.01, 0.001\}$ based on training performance. For CIFAR-10, the initial learning rate is chosen from the set $\{1, 0.1, 0.01, 0.001\}$, the stepsize is divided by 2, 10, 20 after 1000, 3000, 5000 iterations, respectively. For CIFAR-10, the standard deviation of the added noise ($b$ in the algorithm) is set to be different for different weights. Specifically, for weight $W_i$, we set $b = 0.1 * \max_j(Q_{i,j})$ where $Q_{i,j}$ is the maximum of absolute value of elements in stochastic gradient w.r.t $W_i$ at node $j$. This requires transmitting an additional float number per weight but the cost is negligible given the sizes of the weights. We found the aforementioned adaptive noise adding scheme makes it easier to tune the noise level.

## Footnotes

[2] Available at http://yann.lecun.com/exdb/mnist/