[Reviews · NeurIPS 2020]

Review 1

Summary and Contributions: This paper study the convergence properties of Sign and Median SGD algorithm federated learning in heterogeneous data distribution regime. The paper makes several interesting contributions: 1) Making connection between Sign and Median SGD by showing that SignSGD updates along the direction of signed median of gradients. 2) Propose a simple but efficient method for reducing the gap between mean and median statistics. 3) Showing the convergence of Sign and Median SGD in terms of the expected median and mean of gradients at different workers. 4) Using 2 and 3 to propose convergent variant of Sign and Median SGD.

Strengths: - The paper is theoretically sound and very well written. Providing intuitions about each theoretical contribution helps me understand the paper and design choices very well. - The paper is novel and the contributions are significance. Note that heterogeneous data distribution is typically the case in federated learning since each user creates and stores their own data. - I admire the authors for pointing out the superiority of their algorithm in iid setting. It helps the future research understand the problem in more depth. - While the idea of adding noise to reduce the gap between mean and median statistics is intuitive and straightforward to prove it has been not discovered in any other work. I agree with the authors that this could be of independent interest.

Weaknesses: Sign and Median SGD both shows some robustness against Byzantine workers in iid setting. The paper could discuss if similar properties are held in the proposed method when the model is trained on heterogeneous and non-heterogeneous data. After Rebuttal: The author discussed my question regarding the robustness of their method for iid and non-iid data. I hope they add a more detailed discussion in the paper. My overall recommendation is still a clear accept.

Correctness: The method and the experimental protocol seem correct and well designed.

Clarity: The paper is very well written in much details. I appreciate the effort for explaining all the details in a simple language.

Relation to Prior Work: The paper provides a well description of the prior works and their relation to the proposed method.

Reproducibility: Yes

Additional Feedback:


Review 2

Summary and Contributions: In this paper, the authors focus on the problem of median-based distributed training algorithm. The main result is that when the data is draw in an non-IID setting, there is a gap between median-based and mean-based algorithms. And more fascinatingly, this gap can be closed by adding noises to the gradient.

Strengths: - Relevant and important problem. - Interesting result and solid theoretical analysis.

Weaknesses: ~

Correctness: Yes

Clarity: Yes

Relation to Prior Work: Yes

Reproducibility: Yes

Additional Feedback: This is an interesting, solid work and I do not have any major concerns. In my opinion, the problem is relevant and important, and the results are insightful and interesting. One thing that i found interesting is that in Algorithm 3 and 4, the noises added to the gradient is similar to how people achieve differential privacy during training --- since the paper talked about robustness, I wonder what claims the authors can make in terms of privacy --- it seems to me that preserving certain level of privacy is actually helping the convergence for a class of algorithms, improving the utility instead of decreasing it. Maybe this is something to discuss in the paper of having a simple lemma about. ### After rebuttal I liked this paper before rebuttal, and continue to believe that it should be accepted after seeing the author feedback.


Review 3

Summary and Contributions: This paper studies the convergence of sign-SGD and median-SGD under heterogeneous setting. The main conclusion this paper draws is the gap between median and mean will leads to non-convergence. To deal with this issue, a treatment by adding noise is proposed and the convergence has been theoretically justified. Numerical experiments are provided to support the conclusion.

Strengths: Federated learning is really popular recently and the theoretical guarantee is still far from enough. This paper tries to bridge the gap and provides useful intuition.

Weaknesses: My major concern is on the treatment part. Adding noise does make the median closer to mean. However, it also increase the variance.Though the author has carefully studied this in Theorem 6 and shows some trade-off. I still feel this is the most important question to answer here for this treatment.

Correctness: Correct from my side.

Clarity: The paper is overall well written and very easy to read. It has a good introduction part and provide enough intuition to help the readers understand the work.

Relation to Prior Work: Yes.

Reproducibility: Yes

Additional Feedback:

[Author Response · NeurIPS 2020]

We thank the reviewers for the positive feedback. We will answer the questions in the following.

**Reviewer1.** *Q1: Robustness of the proposed methods against Byzantine workers.*

The robustness of Sign and Median SGD both come from the fact that when performing variable updates, the mean of gradients is replaced by the median of gradients, which is less sensitive to extreme values (i.e., those that are possibly provided by Byzantine workers). As noticed by the reviewer, the perturbation mechanism in Section 4 is gradually converting the estimated statistic from median to mean by adding more noise. Thus, the more noise is added, the less robust the estimated statistic will be. This implies that the robustness of Noisy Sign and Median SGD Byzantine workers depends on the the amount of artificial noise ($b$ in Algorithm 3 and 4). On heterogeneous data with Byzantine workers, the performance of the median-based algorithms can be affected by mainly two factors. One is the gap between median and mean, and the other one is the misleading information provided by Byzantine workers. When the possible effect of Byzantine workers is relatively small (e.g. a small number of Byzantine workers) compared with the median-mean gap, some noise is still preferred to reduce the median-mean gap even though this could amplify the effect of Byzantine workers (there could be a trade-off). For non-heterogeneous data (e.g. iid setting), the median could be very close to the mean and the noise scale $b$ in Noisy Sign and Median SGD should be set small or even 0, the effect of such a small noise might be negligible for both convergence and robustness. We will add a more detailed discussion to the paper.

**Reviewer 2.** *Q1: What claims can be made in terms of privacy.*

This is a very interesting question. The noise added to the gradient might provide some level of differential privacy while improving the utility of the algorithms in practice. If one knows the upper bound on $L_2$ norm of all possible gradients, one can use it to calculate the differential privacy cost $\epsilon$ using standard privacy accountant (e.g. Abadi et al. [2016]). However, in many cases the $L_2$ norm bound on gradients is unknown or hard to compute, and gradient clipping is required. It is unclear how the utility of the algorithms will be affected if gradient clipping is applied. We will discuss this question in the paper.

**Reviewer 4.** *Q1: The increased variance in the treatment.*

Indeed, as mentioned by the reviewer, the perturbation mechanism applied in Noisy Sign and Median SGD can increase variance of the original median estimator while making median closer to mean. This mechanism can be understood as trading bias with variance. One needs to properly deal with the added variance in the optimization algorithms. For example, the optimal learning rate of Noisy Sign and Median SGD can depend on the added variance as implied by Theorem 5 and Theorem 6. In practice, such increased variance can change the best learning rate found by hyperparameter search strategies. We will add more discussion on the effect of increased variance.

# References

Martin Abadi, Andy Chu, Ian Goodfellow, H Brendan McMahan, Ilya Mironov, Kunal Talwar, and Li Zhang. Deep learning with differential privacy. In *Proceedings of the 2016 ACM SIGSAC Conference on Computer and Communications Security*, pages 308–318, 2016.


[Meta-Review · NeurIPS 2020]

All reviewers found novelty in the paper, and appreciated the technical contributions on an important problem.